



# Comparing national greenhouse gas budgets reported in UNFCCC inventories against atmospheric inversions

Zhu Deng[1, *], Philippe Ciais[2, *], Zitely A. Tzompa-Sosa[2], Marielle Saunois[2], Chunjing Qiu[2], Chang Tan[1],
Taochun Sun[1], Piyu Ke[1], Yanan Cui[3], Katsumasa Tanaka[2,4], Xin Lin[2], Rona L. Thompson[5], Hanqin Tian[6],
Yuanzhi Yao[6], Yuanyuan Huang[7], Ronny Lauerwald[8], Atul K. Jain[9], Xiaoming Xu[9], Ana Bastos[10],
Stephen Sitch[11], Paul I. Palmer[12,13], Thomas Lauvaux[2], Alexandre d'Aspremont[14], Clément Giron[14],
Antoine Benoit[14], Benjamin Poulter[15], Jinfeng Chang[16], Ana Maria Roxana Petrescu[17], Steven J. Davis[18],
Zhu Liu[1], Giacomo Grassi[19], Clément Albergel[20], and Frédéric Chevallier[2]

[1] Department of Earth System Science, Tsinghua University, Beijing, China
[2] Laboratoire des Sciences du Climat et de l'Environnement/ IPSL, CEA-CNRS-UVSQ, Université Paris-Saclay, 91191 Gif-sur-Yvette, France
[3] Jiangsu Provincial Key Laboratory of Geographic Information Science and Technology, International Institute for Earth System Science, Nanjing University, Nanjing, China
[4] Earth System Risk Analysis Section, Earth System Division, National Institute for Environmental Studies (NIES), Tsukuba, Japan
[5] NILU - Norsk Institutt for Luftforskning, Kjeller, Norway
[6] International Center for Climate and Global Change Research, School of Forestry and Wildlife Sciences, Auburn University, Auburn, AL 36849, USA.
[7] CSIRO Oceans and Atmosphere, Aspendale 3195, Australia
[8] UMR-ECOSYS, Université Paris-Saclay, INRAE, AgroParisTech, Thiverval-Grignon, France.
[9] Department of Atmospheric Sciences, University of Illinois, Urbana, IL 61801, USA
[10] Max Planck Institute for Biogeochemistry, Dept. of Biogeochemical Integration, Hans Knöll Str. 10, 07745 Jena, Germany
[11] College of Life and Environmental Sciences, University of Exeter, Exeter EX4 4RJ, UK
[12] National Centre for Earth Observation, University of Edinburgh, Edinburgh EH9 3FF, UK
[13] School of GeoSciences, University of Edinburgh, Edinburgh EH9 3FF, UK
[14] Kayrros, Paris, France.
[15] NASA GSFC, Biospheric Sciences Lab., Greenbelt, MD 20771
[16] College of Environmental and Resource Sciences, Zhejiang University, Hangzhou, China
[17] Department of Earth and Life Sciences, Faculty of Science, Vrije Universiteit Amsterdam, Amsterdam, The Netherlands
[18] Department of Earth System Science, University of California, Irvine
[19] European Commission, Joint Research Centre, Ispra (VA) Italy.
[20] European Space Agency Climate Office, ECSAT, Harwell Campus, Oxfordshire, Didcot OX11 0FD, UK
[*] These authors contributed equally to this work.

Correspondence: Philippe Ciais (philippe.ciais@cea.fr), Zhu Liu (zhuliu@tsinghua.edu.cn)

**Abstract.** In support of the Global Stocktake of the Paris Agreement on Climate change, this study presents a comprehensive framework to process the results of atmospheric inversions in order to make them suitable for evaluating UNFCCC national inventories of land-use carbon dioxide ($CO_2$) emissions and removals, corresponding to non-fossil sectors. We also deduced anthropogenic methane ($CH_4$) emissions regrouped into fossil and agriculture and waste emissions, and anthropogenic nitrous oxide ($N_2O$) emissions from inversions. To compare inversions with national reports, we compiled a new global harmonized



database of national emissions and removals from periodical UNFCCC inventories by Annex I countries, and from sporadic and less detailed emissions reports by non-Annex I countries, given by National Communications and Biennial Update Reports (no gap filling was applied) . The method to reconcile inversions with inventories is applied to selected large countries covering 78% of the global land carbon uptake for $CO_2$, as well as emissions and removals in the land use, land use change and forestry

sector, and top-emitters of $CH_4$ and $N_2O$. Our method uses results from an ensemble of global inversions produced by the Global Carbon Project for the three greenhouse gases, with ancillary data. We examine the role of $CO_2$ fluxes caused by lateral transfer processes from rivers and from trade in crop and wood products, and the role of carbon uptake in unmanaged lands, both not accounted for by the rules of inventories. Here we show that, despite a large spread across the inversions, the median of available inversion models points to a larger terrestrial carbon sink than inventories over temperate countries or groups of

countries of the Northern Hemisphere like Russia, Canada and the European Union. For $CH_4$, we find good consistency between the inversions assimilating only data from the global in-situ network and those using satellite $CH_4$ retrievals, and a tendency for inversions to diagnose higher $CH_4$ emissions estimates than reported by inventories. In particular, oil and gas extracting countries in Central Asia and the Persian Gulf region tend to systematically report lower emissions compared to those estimated by inversions. For $N_2O$, inversions tend to produce higher anthropogenic emissions than inventories for tropical

countries, even when attempting to consider only managed land emissions. In the inventories of many non-Annex I countries, this can be tentatively attributed to either a lack of reporting indirect $N_2O$ emissions from atmospheric deposition and from leaching to rivers, or to the existence of natural sources intertwined with managed lands, or to an under-estimation of $N_2O$ emission factors for direct agricultural soil emissions. The advantage of inversions is that they provide insights on seasonal and interannual greenhouse gas fluxes anomalies, e.g. during extreme events such as drought or abnormal fire episodes,

whereas inventory methods are established to estimate trends and multi-annual changes. As a much denser sampling of atmospheric $CO_2$ and $CH_4$ concentrations by different satellites coordinated into a global constellation is expected in the coming years, the methodology proposed here to compare inversion results with inventory reports could be applied regularly for monitoring the effectiveness of mitigation policy and progress by countries to meet the objective of their pledges. The dataset constructed by this study is publically available at https://doi.org/10.5281/zenodo.5089799 (Deng et al. 2021).

**Introduction**

Despite the pledges of many countries to limit or decrease their greenhouse gas emissions through the Paris Agreement in 2015, current trends will likely lead to a warming of 3 to 4°C (United Nations Environment Programme, 2020; du Pont and Meinshausen, 2018). Many countries have recently announced ambitious plans to become neutral in terms of their net greenhouse gases emissions in the future, with some ambitious near-term reduction targets. The global stocktake coordinated

by the secretariat of the United Nations Framework Convention on Climate Change (UNFCCC) and the enhanced transparency framework aim to use the best available scientific data and practices for improving national greenhouse gas inventories. Different qualities of inventories are expected among countries (Perugini et al., 2021). UNFCCC Annex I Parties, which

include all OECD countries and several EIT (economies in transition) countries, already have procedures to annually report their emissions in a common reporting format with a time latency of roughly two years. In contrast, non-Annex I Parties,

mostly developing countries and less developed countries, are currently not required to provide reports as regularly and as detailed as Annex I Parties. Only by 2024, one year after the first global stocktake scheduled in 2023, will all countries move to regular reporting of their emissions. Therefore, only non-harmonized data from non-Annex I Parties will be available for the first global stocktake. At the same time, many non-Annex I Parties countries, including the largest emitters, have provided National Communications (NC) or Biennial Update Reports (BUR) to the UNFCCC in the past, so that these data can now be

compared with other estimates. Nevertheless, we note that there are differences in the sectors and sub-sectors of emissions covered by BUR and NCs between non-Annex I inventories, and between Annex I and non-Annex I Parties.

The IPCC guidelines for national greenhouse gas inventories encourage countries to use independent information on emissions and removals, including based on atmospheric concentration measurements interpreted by atmospheric inversion models

(Eggleston et al., 2006; Buendia et al., 2019). Such a verification of 'bottom up' national reports against 'top down' atmospheric inversion results is not mandatory, although a few countries have already added inversions as a consistency check of their national reports (specifically Switzerland (FOEN, 2021), United Kingdom (Brown et al., 2021) , New Zealand (Ministry for the Environment, 2021), and Australia (DISER, 2021)). Here we aim to use the results of available atmospheric inversions with global coverage, focusing on three ensembles of inversions with global coverage published with the global

$CO_2$, $CH_4$ and $N_2O$ budgets assessments coordinated by the Global Carbon Project (GCP) (Friedlingstein et al., 2020; Saunois et al., 2020; Tian et al., 2020). These inversions cover up to the last 40 years for $CO_2$, and the last approximately 20 years for $CH_4$ and $N_2O$.

Inversion results for $CO_2$ land fluxes have been compared with bottom up inventories in previous research work for the USA

(Pacala et al., 2001), geographic Europe (Janssens et al., 2005; Schulze et al., 2009) and China (Piao et al., 2009). Further work was done at the scale of large regions (Canadell et al., 2021). Previously, inversion results were compared to inventories only for one greenhouse gas (Stavert et al., 2020; Thompson et al., 2019), or for one country (Kort et al., 2008; Miller and Michalak, 2017; Miller et al., 2019). Recently, Petrescu et al. (2021a, b) provided a synthesis of the three greenhouse gases emissions over the EU27 + UK for all major emitting sectors in two companion papers using global and regional inversions; the latter

with higher resolution transport models. They also compared UNFCCC inventories with bottom-up datasets: global inventories, vegetation models, forestry models and bookkeeping models analyzing specifically land use change fluxes (as part of the synthesis activity of the VERIFY project) (VERIFY, https://verify.lsce.ipsl.fr/, last access: 01 July, 2021). Yet, for $CO_2$, they did not make any corrections to $CO_2$ inversions for lateral fluxes (Regnier et al., 2013) even though this was done in earlier European syntheses (Ciais et al., 2020b; Janssens et al., 2005).






This study is a contribution to the phase 2 of the REgional Carbon Assessment and Processes (RECCAP2) initiative and takes the next step forward by analyzing UNFCCC inventories for the three greenhouse gases and key sectors, and comparing them with inversions for selected high-emitting countries (or groups of countries) that encompass the majority of global emissions. It also provides detailed methodologies to make inversion results more comparable with inventories, in the context of efforts

made by the scientific community following the roadmap of the Committee on Earth Observation Satellites (The Joint CEOS/CGMS Working Group on Climate, 2020) for using satellite inversions to support the Paris Agreement Global Stocktake process. The methods presented here are also relevant for the development of a global $CO_2$ monitoring and verification support capacity by the European Copernicus Programme (Pinty et al., 2017, 2019; Copernicus, 2021) (e.g. with the CoCO2 project), and by the NASA carbon monitoring system (e.g. the inversions Model Intercomparison of using OCO-

2 satellite data) (NOAA, 2021; Crowell et al., 2019).

The main methodological advances of this study include: 1) the separation of $CO_2$ fluxes from inversions over managed lands in order to better match inventories, 2) the processing of inversion results at the national level to make them more comparable with inventories by subtracting the CO2 fluxes not included in the inventories from their inversions total fluxes, such as $CO_2$

fluxes from lateral carbon transport, 3) the processing of $CH_4$ inversions to split natural and anthropogenic $CH_4$ emissions, enabling the comparison with inventories that only register anthropogenic emissions, 4) a similar treatment of $N_2O$ inversion results, 5) accounting for indirect $N_2O$ emissions from atmospheric anthropogenic nitrogen deposition and man-made nitrogen leaching to groundwater and inland waters, for the countries that did not report these emissions in their inventories. For Annex I countries, annual UNFCCC national inventory report (NIR) data were downloaded from the UNFCCC website (UNFCCC,

2021c) with complete information about each sub-sector, whereas for non-Annex I countries (UNFCCC, 2021a, b), information about sub-sectors is not consistently reported and a manual analysis and quality check of all national communications and biennial update reports had to be done. This new database of all inventory reports contains detailed sub-sectoral information that allows a more accurate re-grouping of emissions into larger sectors, and an easier comparison with inversions.


Our inversion-inventory comparison framework is applicable to countries or groups of countries with an area larger than the spatial resolution of atmospheric transport models typically used for inversions. Further, inversions use a priori information on the spatial patterns of fluxes. Some inversions adjust fluxes at the same spatial resolution of transport models to match atmospheric observations, and use spatial error correlations (usually Gaussian length scales) that tie the adjustment of fluxes

from one grid-cell to its neighbours at distances of hundreds to thousands of kilometers. Other inversions adjust fluxes over coarse regions that are larger than the resolution of parent transport models, implicitly assuming a perfect correlation of fluxes at scales smaller than a coarse region (see Table A4 of Friedlingstein et al. 2020 for $CO_2$ inversions and Table 4 of Saunois et al. 2020 for details). Thus, the results are shown for selected large emitter countries, or large absorbers in the case of $CO_2$. We have selected a different set of countries / groups of countries for each gas. Our selected countries collectively represent 73%

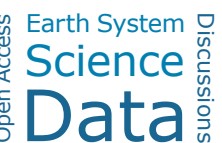

global fossil fuel $CO_2$ emissions, 85% global land $CO_2$ sink, 64% anthropogenic $CH_4$ emissions, and 55% anthropogenic $N_2O$ emissions. To more robustly interpret global inversion results for comparison with inventories, we chose high-emitting countries with an area that contains at least 13 grid boxes of the highest resolution grid-scale inversions and having (if possible) some coverage by atmospheric air-sample measurements, although some selected tropical countries have few or no atmospheric stations (Fig S1). We seek to reconcile inversions with inventories with a clear framework to process inversion

data in order to make them as comparable as possible with inventories. Uncertainties suggested by the spread of different inversion models (min-max range given the small number of inversions), and the causes for discrepancies with inventories are analyzed systematically and on a case-by-case basis, for annual variations and for mean budgets over several years. We specifically address the following questions: 1) how do UNFCCC emission inventories compare with inversions for each of the three gases?; 2) what are plausible reasons for mismatches between inversions and UNFCCC inventories?; 3) what

independent information can be extracted from inversions to evaluate the mean values or the trends of greenhouse gas emissions and removals?; 4) can inversions be used to constrain national $CH_4$ emissions and separate trends for natural versus anthropogenic sources, and into fossil fuels and agricultural plus waste anthropogenic sources?; and 5) can inversions help constrain the importance of natural emissions and removals (not reported by inventories, and yet important for linking emissions and removals to the concentration and radiative forcing changes) of the three main greenhouse gases?


The paper presents a new global database of national emissions reports for all countries and its grouping into sectors, the global atmospheric inversions used for the study, the processing of fluxes from these inversions to make their results as comparable as possible with inventories (section 1), the time series of inversions compared with inventories for each gas, with insights on key sectors for $CH_4$ (sections 2-4). The discussion (section 5) focuses on the comparison of terrestrial $CO_2$ fluxes with fossil

and cement emissions, the different sources of uncertainties for $CH_4$ inversions, the comparison of inversions with $CH_4$ and $N_2O$ inventories for mean budgets in the most recent 5-years, and the comparison of global inversion results from this study with published regional inversions. Finally, concluding remarks are drawn on how inversions could be used in a systematic manner to support the evaluation and possible improvement of inventories for the Paris Agreement.

## 1 Material and methods

### 1.1 Compilation and harmonization of national inventories reported to the UNFCCC

All UNFCCC Parties "shall" periodically update and submit their national GHG inventories of emissions by sources and removals by sinks to the Convention parties. Annex I countries have submitted their national inventory reports (NIRs) and common reporting format (CRF) tables every year with a complete time series starting in 1990. Non-Annex I Parties have been required to submit their national communications (NC) roughly every four years after entering the Convention, and

submit Biennial Update Reports (BUR), every two years since 2014. Currently, there are in total nearly 400 submissions of



NC and over 100 submissions of BUR. The non-Annex I reports are in PDF format only, while Annex-I countries reports are provided as Excel files under a Common Reporting Format (CRF) (Fig 1).

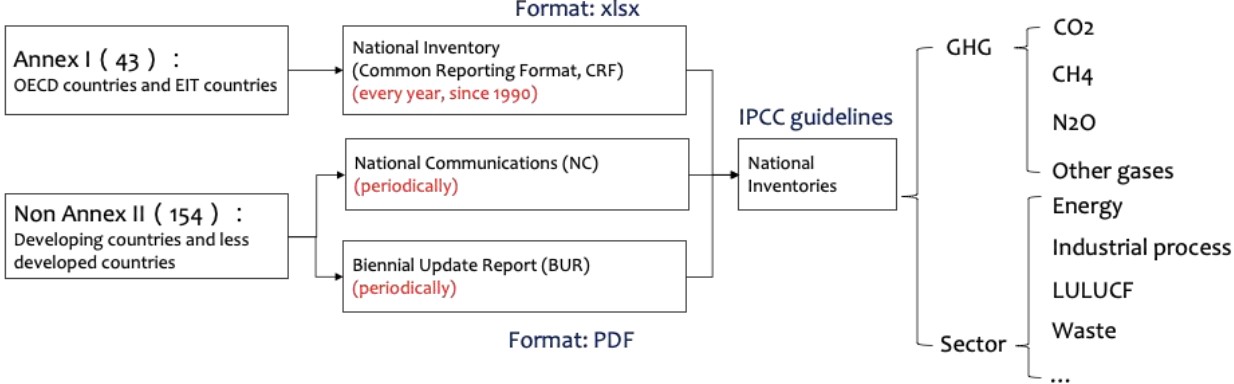

**Figure 1. Structure of national inventory submissions from Annex I and non-Annex I countries.**


We collected the greenhouse gas emission data from the national inventories submitted to UNFCCC. For Annex I countries, data collection is straight-forward, as the common reporting format is computer readable. For non-Annex I countries, the data were directly extracted from the original reports provided in Portable Document Format (PDF). Data from successive reports for the same country were extracted, except when they relate to the same years, in which case only the latest version is

considered. The Annex I countries are required to compile their inventory following a classification into sectors. The reported sectors are different among NC and BUR reports, mainly corresponding to IPCC 1996 or IPCC 2006 definitions.

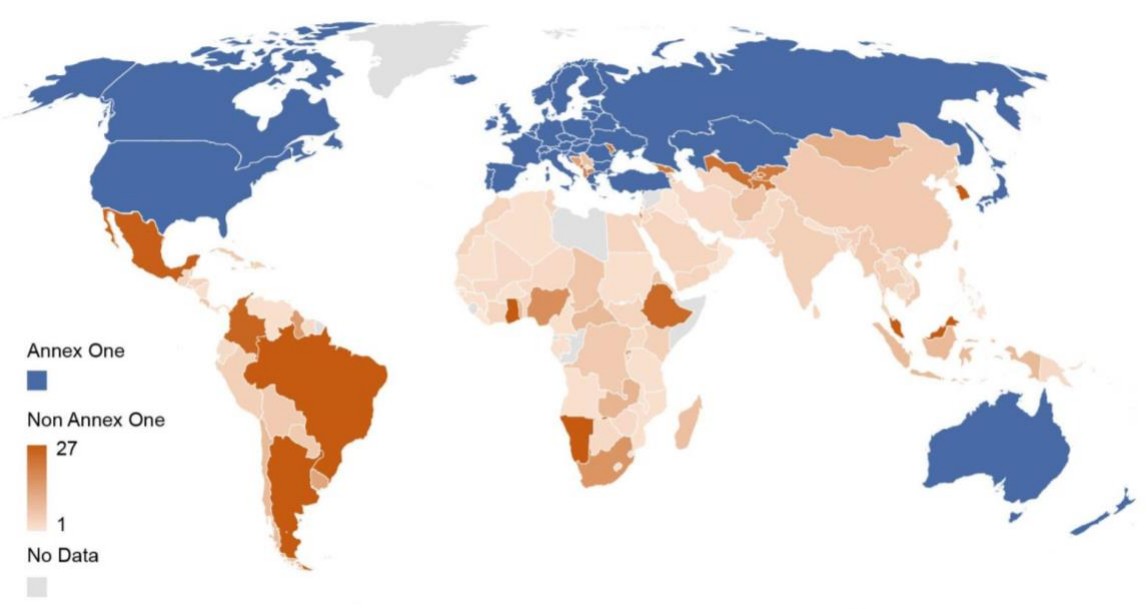



**Figure 2. Numbers of years covered by national inventories reports (NC+BUR) in each non-Annex I country; Emissions from Greenland are reported by Denmark.**


## 1.2 Atmospheric inversions

### CO₂ inversions

The $CO_2$ atmospheric inversions used here (Table S1) are the six from the Global Carbon Budget 2020 (Friedlingstein et al., 2020): CarbonTracker-Europe CTE2020 (van der Laan-Luijkx et al., 2017), the Jena Carboscope sEXTocNEET_v2020

(Rödenbeck et al., 2003), the inversion from the Copernicus Atmosphere Monitoring Service (CAMS) (Chevallier et al., 2005) v19r1, the inversion from the University of Edinburgh (Feng et al., 2016), the Model for Interdisciplinary Research on Climate inversion (MIROC) (Patra et al., 2018) and the NICAM-based Inverse Simulation for Monitoring CO2 (NISMON-CO2) v2020.1 (Niwa, 2020; Niwa et al., 2017b). They all cover at least the period 2001-2019 based on atmospheric air-sample measurements. Their design is summarized in Tables 4 and A4 of Friedlingstein et al. (2020). A common protocol unites them

but this protocol only deals with the submission procedure and data formats: participants were free to design their inversion configuration in their own way, as long as their resulting inversion satisfied some quality criterion. A common gridded fossil fuel dataset with monthly resolution (Jones et al., 2021) was made available to the participants as a fixed prior, but its use was not compulsory.

### CH₄ inversions

The $CH_4$ atmospheric inversions used here (Table S1) to estimate methane fluxes are those from eight inverse systems reporting for the Global Methane Budget (Saunois et al., 2020): CarbonTracker-Europe CH4 (Tsuruta et al., 2017), GELCA (Ishizawa et al., 2016), LMDZ-PYVAR (Yin et al., 2015; Zheng et al., 2018a, b), MIROC4-ACTM (Patra et al., 2018; Chandra et al., 2021), NICAM-TM (Niwa et al., 2017a, b), NIES-TM-FLEXPART (Wang et al., 2019; Maksyutov et al., 2021), TM5-CAMS (Segers and Houweling, 2017) and TM5-JRC (Bergamaschi et al., 2018). An ensemble of 21 inversions includes 10 surface-

based inversions covering 2000-2017 and 11 satellite-based inversions covering 2010-2017 (Table S1). The protocol suggested a set of common prior source and sink estimates along with a set of in-situ atmospheric observations. However, their use was not compulsory, and the inversions differ in terms of prior fluxes and handling of observation data. Satellite-based inversion uses TANSO-GOSAT $CH_4$ total columns, but different retrievals were used depending on the modelling group (see Saunois et al., 2020 supplementary material). As a result, the ensemble of $CH_4$ inversions derived a wider range of results compared to

those from a strict inter-comparison protocol. However, most of the inversions were driven using prescribed climatological OH from TRANSCOM (Patra et al., 2011). Omitting OH interannual variability and trends leads to attributing most of the variations in atmospheric methane concentration to variations in emissions.



**N₂O inversions**

The N₂O atmospheric inversions used to estimate N₂O fluxes are the three inversion systems used in the GCP Nitrous Oxide
Budget (Tian et al., 2020): GEOS-Chem (Wells et al., 2015), PyVAR-CAMS (Thompson et al., 2014), and INVICAT (Wilson et al., 2014). The MIROC4-ACTM N₂O inversion was not used as it has a relatively coarse-resolution control vector and appears to be an outlier (Patra et al., 2018). Similarly to CH₄, the protocol recommended a set of prior source and sink estimates but these were not compulsory, although all the three inversions used in this study used the same prior estimates. All inversions used ground-based observations from the NOAA discrete sampling network, and three of the inversions included observations
from additional networks (for details see Tian et al., 2020). All inversions accounted for photolysis and oxidation of N₂O in the stratosphere resulting in atmospheric lifetimes in the range of 118 to 129 years.

**1.3 Processing of CO₂ inversions data for comparison with inventories**

**National masks - fossil fuel emissions regridding - managed land mask**

The aggregation of the gridded flux maps of each inversion, with various native resolutions, at the national annual scale followed the procedure described in Chevallier (2021): it was based on the 0.08°×0.08° land country mask of Goldewijk et al. (2017) that allowed us to compute the fraction of each country in each inversion grid box. In addition, for CH₄ and N₂O, emissions from inland waters at 0.08°×0.08° resolution were attributed to the closest country. For this study, intact forest areas (that are unmanaged, by definition) were removed from the CO₂ totals, in proportion to their presence in each inversion grid
box, based on the Intact Forest Landscapes maps of Potapov et al. (2017) shown in Figure S1. This approach assumes that non-intact forest represents a reasonably good proxy of managed forest reported in national GHG inventories (Grassi et al., 2021). In the absence of a machine-readable definition of the plots considered to be managed in many NIRs, this choice remains somewhat arbitrary and other unmanaged land datasets could have been used (Ogle et al., 2018; Chevallier, 2021). We subtracted the same fossil fuel emissions from Friedlingstein et al. (2020) from the total CO₂ flux of each inversion to analyze
terrestrial CO₂ fluxes, which is equivalent to assuming perfect knowledge of emissions but note that these values are consistent with the fossil fuel emissions reported in the NIRs. This assumption leads to an under-estimation of the spread of terrestrial CO₂ fluxes among inversions.

**Subtracting CO₂ fluxes due to lateral carbon transport by crop and wood products trade and by rivers**

We subtracted from inversions' national estimates of the Net Ecosystem Exchange (NEE) CO₂ flux ($F^{inv\,NEE}_{ML}$) over managed
lands (*ML* here defined as all land except intact forests), a set of CO₂ fluxes which are not counted by NIs. Here we use NEE from the definition of Ciais et al. (2020b), for all non-fossil CO₂ exchange fluxes between terrestrial surfaces and the atmosphere; other work may use Net Biome Production (NBP) with a similar meaning. As a result, we produce 'adjusted' inversion fluxes that can be compared to inventories. The CO₂ fluxes that are part of $F^{inv\,NEE}_{ML}$ but are not counted by NIs, are





induced by (i) anthropogenic export and import of crop and wood products across each country's boundary ($F_{ant}^{crop\ trade}$ and

$F_{antt}^{wood\ trade}$), and (ii) river carbon export ($F_{tot}^{rivers}$) which has an anthropogenic and a natural component (Regnier et al., 2013).

We assumed that NIs include losses from fire (wildfire and prescribed fire) and other disturbances (wind, pests) and from

harvesting in their estimates of land carbon stocks changes, as recommended by the LULUCF reporting guidelines. The

adjusted inversion NEE that can be compared with inventories, $F_{adj}^{inv\ NEE}$, is given by:

$$F_{adj}^{inv\ NEE} = F_{ML}^{inv\ NEE} - F_{tot}^{rivers} - F_{ant}^{crop\ trade} - F_{ant}^{wood\ trade} \quad \Leftrightarrow \quad F_{ant}^{ni},  \tag{1}$$

where the sign ⇔ means 'compared with', $F_{tot}^{rivers}$ is the sum of natural and anthropogenic $CO_2$ uptake flux on land from $CO_2$

fixation by plants that is leached as carbon via soils and channeled to rivers to be exported to the ocean or to another country.

All countries export river carbon, but some countries also receive river inputs, e.g. Romania receives carbon from Serbia via

the Danube river. We estimated the lateral carbon export by rivers minus the imports from rivers entering in each country,

including dissolved organic carbon, particulate organic carbon and dissolved inorganic carbon of atmospheric origin

distinguished from lithogenic, by using the data and methodology described by Ciais et al. (2020b). Data are from Mayorga et

al. (2010) and Hartmann et al. (2009) and follow the approach of Ciais et al. (2020a, b) proposed for large regions, but here

with new data at national scale. Over a country that only exports river carbon, the amount of carbon exported is equivalent to

an atmospheric $CO_2$ sink, denoted as $F_{tot}^{rivers}$ as in eq. (1), thus ignoring burial, which is a small term. Over a country that

receives carbon from rivers flowing into its territory, a small national $CO_2$ outgassing is produced by a fraction of this imported

flux. In that case, we assumed that the fraction of outgassed to incoming river carbon is equal to the fraction of outgassed to

soil-leached carbon in the RECCAP2 region to which a country belongs to, estimated with data from Ciais et al. (2020b).

$F_{ant}^{crop\ trade}$ is the sum of $CO_2$ sinks and sources induced by the trade of crop products. This flux was estimated from the annual

trade balance of 171 crop commodities calculated for each country from FAOSTAT data (FAOSTAT, 2021) combined with

carbon content values of each commodity (Xu et al., 2021). All the traded carbon in crop commodities is assumed to be

oxidized as $CO_2$ in one year, neglecting stock changes of products, and the fraction of carbon from crop products going to

waste pools and sewage waters after consumption, thus not necessarily oxidized to atmospheric $CO_2$. $F_{ant}^{wood\ trade}$ is the sum

of $CO_2$ sinks and sources induced by the trade of wood products (Zscheischler et al., 2017). Here, we followed Ciais et al.

(2020b) who used a bookkeeping model to calculate the fraction of imported carbon in wood products that is oxidized in each

country during subsequent years, defined from Mason Earles et al (2012). Emissions of $CO_2$ by herbivory is partly included in

the $F_{ant}^{crop\ trade}$ flux for the fraction of crop products delivered as feed to animals. Emissions of $CO_2$ from grazing animals occur

in the same grid box where grass is consumed, so that the $CO_2$ net flux captured by an inversion is comparable with grazed

grasslands carbon stock changes of inventories. Emissions of reduced carbon compounds (VOCs, $CH_4$, CO) are not included

in this analysis (see Ciais et al. 2020b for discussion of their importance in inversion $CO_2$ budgets).

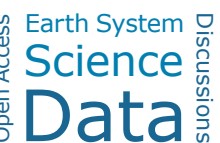

In summary, the purpose of the adjustment of equation (1) is to make inversions output comparable to the inventories that do not include $F_{tot}^{rivers}$, $F_{ant}^{crop\ trade}$ and $F_{ant}^{wood\ trade}$. For example, the UNFCCC accounting rules assume that all the harvested

wood products are emitted in the territory of a country which produces them, which is equivalent to ignoring $F_{ant}^{wood\ trade}$ as a national sink or source of $CO_2$. The adjusted inversion fluxes from equation (1) no longer correspond to physical real land-atmosphere $CO_2$ world-fluxes, but they match the carbon accounting system boundaries of UNFCCC inventories and will be used in the following. In the following, we will only discuss adjusted inversion $CO_2$ fluxes, but for simplicity call them "inversion fluxes".


**1.4 Processing of CH₄ inversions for comparison with national inventories**

Atmospheric inversions usually derive net $CH_4$ emissions at the surface as it is difficult for them to disentangle overlapping emissions from different sectors at the pixel/regional scale based on atmospheric $CH_4$ observations only. However, six of the eight modelling systems solve for some source categories owing different spatio-temporal distributions between the sectors.

For each inversion, monthly gridded posterior flux estimates were provided at 1°x1° grid resolution for the net flux at the surface ($E_{net}^{inv}$), the soil uptake at the surface ($E_{soil}^{inv}$), the total source at the surface ($E_{tot}^{inv}$) and five emitting sectors: Agriculture & Waste ($E_{AgW}^{inv}$), Fossil Fuel ($E_{FF}^{inv}$), Biomass & Biofuel Burning ($E_{BB}^{inv}$), Wetlands ($E_{Wet}^{inv}$), and Other Natural ($E_{Oth}^{inv}$) emissions. Considering the soil uptake as a 'negative source', the following equations apply:

$$E_{net}^{inv} = E_{tot}^{inv} + E_{soil}^{inv} = E_{AgW}^{inv} + E_{FF}^{inv} + E_{BB}^{inv} + E_{Wet}^{inv} + E_{Oth}^{inv} \tag{2}$$

For inversions solving for net flux only, the partition to source sectors was created based on using a fixed ratio of sources calculated from prior flux information at the pixel scale. For inversions solving for some categories, a similar approach was used to partition the solved categories to the five aforementioned emitting sectors. Such processing can lead to significant

uncertainties if not all sources increase or change at the same rate in a given region/pixel. National values have been estimated using the country land mask described in the $CO_2$ section, thus offshore emissions are not counted as part of inversion results unless they are in a coastal grid-cell.

Four methodologies were used to separate $CH_4$ anthropogenic emissions from inversions ($E_{Anth}^{inv}$) in order to compare them

with national inventories ($E_{Anth}^{ni}$). The first method consists in using the inversion partitioning as defined in Saunois et al. (2020):

**Method 1**



$$E_{Anth}^{inv} = E_{AgW}^{inv} + E_{FF}^{inv} + E_{BB}^{inv} - E_{wildfires}^{BU} \Leftrightarrow E_{Anth}^{ni} \qquad (3)$$

This method has some uncertainties. First, the partitioning relies on the prior estimates, and second, emissions from wildfires are counted for in the Biomass and Biofuel burning (BB) inversion category while they are not reported in national inventories. The BB inversion category includes methane emissions from wildfires in forests, savannahs, grasslands, peats, agricultural residues, and the burning of biofuels in the residential sector (stoves, boilers, fireplaces). Therefore, we subtracted bottom-up (BU) emissions from wildfires ($E_{wildfires}^{BU}$) based on the GFEDv4 dataset (van der Werf et al., 2017) using their reported dry matter burned and $CH_4$ emission factors. Because the GFEDv4 dataset also reports specific agricultural and waste fires emissions data, and we assume that those fires (on managed lands) are reported by NIs, they were not counted in $E_{wildfires}^{BU}$.

**Method 2 and 3**

The second method is a variant of the first one, which removes the median of all inversions of natural emissions (wetlands and other natural sources in Saunois et al., 2020) from the total sources though the bias related to the *BB* category still exists in this method.

The third method removes natural emissions using products from bottom-up approaches. These methods rely first on the soil uptake, either prescribed or optimized, in the inversion, in order to determine the total methane emissions (anthropogenic+natural).

$$E_{net}^{inv} - E_{soil}^{inv} = E_{tot}^{inv} = E_{Ant}^{inv} + E_{Nat}^{inv} \qquad (4)$$
$$E_{Ant}^{inv} = E_{tot}^{inv} - E_{Ter}^{BU} - E_{Wet}^{BU} - E_{Fre}^{BU} - E_{Geo}^{BU} - E_{wildfires}^{BU} \Leftrightarrow E_{ant}^{ni} \qquad (5)$$

The natural methane sources removed from total $CH_4$ emissions in Eq. (5) are from bottom up (BU) emission estimates from termites $E_{Ter}^{BU}$, wetlands $E_{Wet}^{BU}$, freshwater (lakes and reservoirs) $E_{Fre}^{BU}$, and geological processes $E_{Geo}^{BU}$. Termites emissions are described in Saunois et al. (2020) and we use the mean of the estimates that amounts to 9 TgCH4 yr-1 at the global scale. Geological emissions are based on the Etiope et al. (2019) distributions with a global initial total of 37.4 TgCH4 yr-1 rescaled to a lower value of 5.4 TgCH4 yr-1 in agreement with pre industrial radiocarbon-CH4 measurements of (Hmiel et al., 2020). Freshwater emissions from lakes and reservoirs are from Stavert et al. (2020) and contribute about 71.6 Tg CH4 yr-1 at the global scale, likely an overestimation due to double counting with wetlands (Saunois et al., 2020). It should be noted that fluxes with the inland water surfaces are attributed to the closest country by using the high-resolution country mask described in the $CO_2$ section to avoid double counting. Two variants of the third method were used, differing by the bottom-up product used



to remove wetland emissions. In method 3/1, we use a climatological estimate of wetland emissions calculated from land surface models forced by the same wetland extent WAD2M (Zhang et, 2021) and in method 3/2 we use the emissions of the same land surface models simulating variable wetland areas. The calculations of anthropogenic emissions by each method was

performed separately for GOSAT inversions and in-situ inversions.

**1.5 Processing of N₂O inversions for comparison with inventories**

We subtracted estimates of natural N₂O sources from the N₂O emission budget ($E_{tot}^{inv}$) of each inversion, in order to provide inversions of anthropogenic emissions ($E_{ant}^{inv}$) that can be compared with national inventories ($E_{ant}^{ni}$).


$$E_{ant}^{inv}= E_{managed\ land}^{inv} - E_{nat}^{aq} - E_{wildfires}^{GFED} \Leftrightarrow E_{ant}^{ni} \tag{6}$$

For this study, intact forest areas (that are unmanaged, by definition) from Potapov et al. (2017) and lightly grazed grassland areas from Chang et al. (2021a) were removed from the N₂O totals in proportion to their presence in each inversion grid box.

Lightly grazed grasslands (Chang et al., 2021a) include ecosystems with wild grazers and with extensive grazing by domestic animals, mainly in steppe and tundra regions (Fig S1). We consider that these unmanaged systems emit N₂O from natural processes, but inventories do not apply any specific emission factor for them and assume zero emissions. We verified that the inversion grid boxes fractions classified as unmanaged do not contain point source emissions from the industry, energy and diffuse emissions from the waste sector, to make sure that we do not inadvertently remove anthropogenic sources by masking

those unmanaged areas.

From the EDGARv4.3.2 inventory (Janssens-Maenhout et al., 2019), we found that N₂O from waste water handling covers a relatively large area that might be partly located in unmanaged land. But the emission rates are more than 1 order of magnitude smaller than from agriculture soils. For other sectors, only very few of the unmanaged grid boxes contain point sources, and none of them has an emission rate that is comparable with agricultural soils (managed land). Thus, our assumption that

emissions from these other sectors are primarily from managed land is solid (other sectors include: power industry; oil refineries and transformation industry; combustion for manufacturing; aviation; road transportation no resuspension; railways, pipelines, off-road transport; shipping; energy for buildings; chemical processes; solvents and products use; solid waste incineration; waste water handling; solid waste landfills). The flux $E_{nat}^{aq}$ is the natural emission from freshwater systems given

by a gridded simulation of the DLEM model (Yao et al., 2019) describing pre-industrial N₂O emissions from N leached by soils and lost to the atmosphere by rivers in absence of anthropogenic perturbations (considered as the average of 1900-1910). Natural emissions from lakes were estimated only at global scale by Tian et al. (2020), and represent a small fraction of rivers' emissions. Therefore, they are neglected in this study. The flux $E_{wildfires}^{GFED}$ is based on the GFED4s dataset (van der Werf et

al., 2017) using their reported dry matter burned and $N_2O$ emission factors. Because the GFED dataset reports specific

agricultural and waste fires emissions data, and we assume that those fires (on managed lands) are reported by NIs, so that they were not counted in $E^{GFED}_{wildfires}$. Note that there could also be a background natural $N_2O$ emission from soils over managed lands ($E^{soil}_{managed\ land}$). We did not try to subtract this flux from managed land emissions because we assumed that, after a land use change from natural to fertilized agricultural land, background emissions decrease and become very small compared to N-fertilizers induced anthropogenic emissions. In a future study, we could use for $E^{soil}_{managed\ land}$ the estimate given by

simulatations of pre-industrial emissions from the NMIP ensemble of dynamic vegetation models with carbon-nitrogen interactions (number of models; n = 7). Namely, their simulation S0 in which climate forcing is recycled from 1901-1920; $CO_2$ is at the level of 1860, and no anthropogenic nitrogen is added to terrestrial ecosystems (Tian et al., 2019).

Another important point to ensure a rigorous comparison between inversions and NI data is whether anthropogenic indirect

emissions (AIE) of $N_2O$ are counted in NI reports. Those AIE arise from anthropogenic nitrogen from fertilizers leached to rivers and anthropogenic nitrogen deposited from the atmosphere to soils. AIEs represent typically 20% of the direct anthropogenic emissions and cannot be ignored in a comparison with inversions. For Annex I countries, AIEs are systematically reported, generally based on ad-hoc emission factors since these fluxes cannot be directly measured, and it is assumed that indirect emissions only occur on managed land. For non-Annex I countries, the UNFCCC website gives $N_2O$

emissions for the energy, industry, agriculture, waste and other sectors, with sub-sectors details, without details about AIE. We thus checked manually from the original NC and BUR documents if AIE were reported or not by each non-Annex I country. If AIEs were reported by a country, they were used as such to compare NI data with inversion results, and grouped into the agricultural sector. If they were not reported, or if their values were outside plausible ranges, AIE were independently estimated by the perturbation simulation of N fertilizers leaching, $CO_2$ and climate on rivers and lakes fluxes in the DLEM

model (Yao et al., 2019), and by the perturbation simulation of atmospheric nitrogen deposition on $N_2O$ fluxes from the NMIP model ensemble (Tian et al., 2019). Table S2 lists the non-Annex I countries among the top-20 $N_2O$ emitters whether they have reported AIE to UNFCCC from national inventories.

**1.6 Grouping of inventory sectors for comparison with inversions sectors**

The classification of inversion sectors and inventory sectors is different. The bottom-up inventories are compiled based on activity data (statistics) by following the IPCC 1996 Guidelines (Houghton et al. 1997) or IPCC 2006 Guidelines (Eggleston et al. 2006) for National Greenhouse Gas Inventories, with detailed information of subsectors. But the top-down inversions can only distinguish very few sectors. Thus, in this study, we group the sectors into some aggregated sectors to make inversions and inventories comparable for each GHG gas (Table 1).






For $CO_2$, the inversions are divided into two aggregated sectors: fossil fuel and cement $CO_2$ emissions, and land flux. Inversions use a prior gridded fossil fuel dataset as summarized in Section 1.2, thus, in this study we compare the land flux between inversions and inventories. The land flux (NEE) from each inversion is calculated by subtracting the national total fossil emission from the total $CO_2$ flux, where the fossil emissions are from the Global Carbon Project annual dataset (Friedlingstein

et al., 2020) consistent with the prior fossil emissions maps proposed to inversion modellers and used by them, except for the inversion of Feng et al. (2009; 2016) (see table A4 of Friedlingstein et al. 2020 for details). For inventories, we removed from the total reported $CO_2$ flux the fossil emissions of the sectors of 'Energy' and 'Industrial Processes' to obtain terrestrial $CO_2$ fluxes over managed ecosystems. Note that transportation and residential $CO_2$ emissions are reported under the Energy sector.

For $CH_4$, we compare inversions and national inventories based on three emission groups: fossil sector, agriculture and waste sector, and total anthropogenic sector. For national inventories, the fossil sector includes the categories 'Energy' and 'Industrial Processes' but excludes 'Biofuel Burning' (reported under 'Energy' sector) to better match the inversion categories; and total anthropogenic emissions includes 'Energy', 'Industrial Processes', 'Agriculture' and 'Waste'. Agriculture and Waste emissions are grouped to be compared to 'Agriculture & Waste' from the inversions.


For $N_2O$, we derived anthropogenic emissions by equation (6) by subtracting natural emissions from rivers from (Yao et al., 2019) after masking unmanaged grasslands and intact forest areas.

**Table 1. Grouping of aggregated sectors for comparisons between inventories and inversions. * Biofuel burning is likely**
**not included in NIs but under *1.A.4 Other Sectors* if it is reported. ** Field burning of agricultural residues is reported in Annex I countries under the Agricultural sector. Note that indirect $N_2O$ emissions are reported by Annex I countries but not systematically by non-Annex I ones (see Table S2)**

| Gas | Aggregated sectors in this study | Inversions | Inventories |
|---|---|---|---|
| $CO_2$ | Net Land Flux | Total - Fossil | Net emissions - (Energy + Industrial Processes) |
| $CH_4$ | Total Anthropogenic | Fossil + Agriculture & Waste + Biomass Burning | Energy + Industrial Processes + Agriculture + Waste + Biomass Burning |
| | Fossil (including oil, gas, coal) | Fossil | Energy + Industrial Processes - Biofuel Burning* |
| | Agriculture & Waste | Agriculture & Waste | Agriculture + Waste - Field burning of agricultural residues** |
| $N_2O$ | Anthropogenic | Total - pre-industrial | Agriculture + Waste direct + anthropogenic |



| inland waters - pre-industrial soil emissions | indirect emissions (AIE = anthropogenic N leached to inland waters + anthropogenic N deposited from atmosphere) + energy and industry |

### 1.7 Choice of example countries for analysis

We selected 12 countries for analysis, the selection being different for $CO_2$, $CH_4$ and $N_2O$ anthropogenic fluxes, based on the following criteria. Each selected country should have a large enough area, because small countries cannot be constrained using coarse resolution inversions, and if possible some coverage by the in-situ global network. The country with the smallest area is Venezuela (916,400 km$^2$), selected for $CH_4$ because it is a large oil and gas emitter, and its emissions can still be constrained by inversions using GOSAT satellite observations, excepted inversions using the NIES column $CH_4$ product that has very few

observations in the wet season over Venezuela and Nigeria for instance (see Table S1 and Fig S2 for GOSAT satellite soundings coverage). For $CO_2$, we selected the top ten fossil fuel $CO_2$ emitters, because even if inversions do not resolve those emissions which are used as fixed prior, it is important to compare the magnitude of their $CO_2$ sinks with their $CO_2$ emissions. We also selected two large boreal countries (i.e., Russia and Canada), two tropical countries with important areas of forests (i.e., Brazil and Democratic Republic of Congo), two large countries with in-situ stations (i.e., Mongolia and Kazakhstan), and

two large dry southern countries with a high rank in the fossil fuel $CO_2$ emitters (i.e., South Africa and Australia) which both have atmospheric stations to constrain their land $CO_2$ flux. Altogether, the 12 countries account for 23% of the global land $CO_2$ sink given by NIs. For $CH_4$, we ranked countries by decreasing order of total anthropogenic, fossil and agricultural emissions. The criteria of large areas and having atmospheric stations is important for in-situ inversions. For satellite inversions, the advantage of GOSAT is that it provides observations where the surface network is very sparse, such as the

tropics, so that countries with no or only a few ground-based observations can still get reliable top-down estimates. The inversion resolution is what dictates if small countries can be reliably estimated. Thus, this study includes China, India, USA, EU, Russia, Indonesia, which are among the top fossil and top agricultural emitters and are with vast territory, except for the small countries considering the coarse spatial resolution. Altogether, the selected countries for $CH_4$ represent 57% of the global anthropogenic $CH_4$ emission given by NIs, 55% of the fossil emission and 54% of agriculture and waste emissions. For $N_2O$,

we chose the top 12 emitters based on NI reports. In most of them, anthropogenic $N_2O$ emissions are dominated by the agricultural sector, whose share (including indirect agricultural emissions) to total NI emissions ranges from 6% in Venezuela to 95% in Brazil. Altogether, the selected countries represent 54% of the global anthropogenic $N_2O$ emissions given by NIs.

**Table 2. Lists of countries or groups of countries analyzed and displayed in the result section for each gas and each**

**sector: China (CHN), United States (USA), Russia (RUS), Canada (CAN), Kazakhstan (KAZ), Mongolia (MNG), India (IND), BRA (Brazil), COD (Democratic Republic of the Congo), South Africa (ZAF), Australia (AUS), EU27 & UK**



**(EUR) = EU27 + the United Kingdom, Pakistan (PAK), Argentina (ARG), Mexico (MEX), Iran (IRN), Indonesia (IDN), GULF = Saudi Arabia + Oman + United Arab Emirates + Kuwait + Bahrain + Iraq + Qatar, KT = Kazakhstan + Turkmenistan, Venezuela (VEN), Nigeria (NGA), Thailand (THA), Bangladesh (BGD), Columbia (COL), Sudan (SDN).**

| Gas | Sector | Country List |
|---|---|---|
| $CO_2$ | Net Land Flux | AUS, BRA, CAN, CHN, COD, EUR, IND, KAZ, MNG, RUS, USA, ZAF |
| $CH_4$ | Anthropogenic | ARG, AUS, BRA, CHN, EUR, IDN, IND, IRN, MEX, PAK, RUS, USA |
| | Fossil (including oil, gas, coal) | CHN, EUR, GULF, IDN, IND, IRN, KT, MEX, NGA, RUS, USA, VEN |
| | Agriculture & waste | ARG, BGD, BRA, CHN, EUR, IDN, IND, MEX, PAK, RUS, THA, USA |
| $N_2O$ | Anthropogenic | AUS, BRA, CHN, COD, COL, EUR, IDN, IND, MEX, SDN, USA, VEN |

## 2 Results for net land $CO_2$ fluxes

First, we compared the global land $CO_2$ sink from inversions with inventories. The data compiled by Grassi et al. (2021) include Annex I countries reports, non-Annex I NC and BUR and FAOSTAT estimates for some non-Annex I countries. We took the average of Grassi et al. (2021)for 2010 and 2015. Inversion median values were calculated for the period 2007-2017 that roughly cover these two years. Inversions give an average global land sink of 2.5 GtC yr$^{-1}$ over all lands and of 2.7 GtC yr$^{-1}$ over managed lands. Over managed land, the CAMS, UoE and Jena models have a higher sink, CTE and NISMON give similar values, and MIROC gives a smaller sink than over all lands (See Table S1 for the list of models). In contrast, inventory data compiled by Grassi et al. (2021) indicates a global land sink of only 0.3 GtC yr$^{-1}$. Such a large difference can be possibly explained by the fact that NIs are incomplete (especially in developing countries, and especially non-forest land uses such as cropland, grassland and wetlands) and do not fully capture recent environmental and meteorological effects, such as the impact of more frequent extreme weather events (droughts), and possibly by the lack of actual observation-based estimates in inventories to constrain soil carbon change, in grasslands, croplands and forests. Inversions are also larger than the Tier-1 forest inventory recently published by Harris et al. (2021), who estimates a sink of 2.1 GtC yr$^{-1}$ over the last 20 years with a range of ±13 GtC, which seems biophysically implausible.

Figure 3 displays the time series of land-to-atmosphere $CO_2$ fluxes for the selected countries (Table 2). Across the 12 countries, the median of inversions shows significant interannual variability, generally consistent between the six inversions (Fig S3).



This signal reflects the impact of climate variability on terrestrial carbon fluxes, and annual variations of land-use emissions.

In the inversion of $CO_2$ fluxes, the effects of climate variability on an inter-annual and decadal scale, rising $CO_2$, nitrogen availability and other environmental drivers are not separable from the direct human-induced effects of land use and management. Decadal variability of carbon stocks induced by climate and environmental drivers is mostly captured by the NI of countries that use regular forest inventories to measure stock changes over time (stock-difference method), e.g. with dense sampling of forest plots. Yet, such gridded stock change inventories do not capture interannual variability, for instance, when

higher mortality or a growth deficit occurs in a severe drought year and causes an abnormal $CO_2$ source to the atmosphere (Ciais et al., 2005; Wolf et al., 2016; Bastos et al., 2020). In contrast, the NIs of countries based on forestry models using static growth curves of representative forests do not necessarily capture the recent transient impact of environmental driver changes, see Supp. Table 1 of Grassi et al. (2021) which includes information on the method (gain-loss or stock difference) used by several major countries. This may partially explain why inversions estimate higher $CO_2$ sinks (e.g. in CAN, AUS and some

EUR countries).

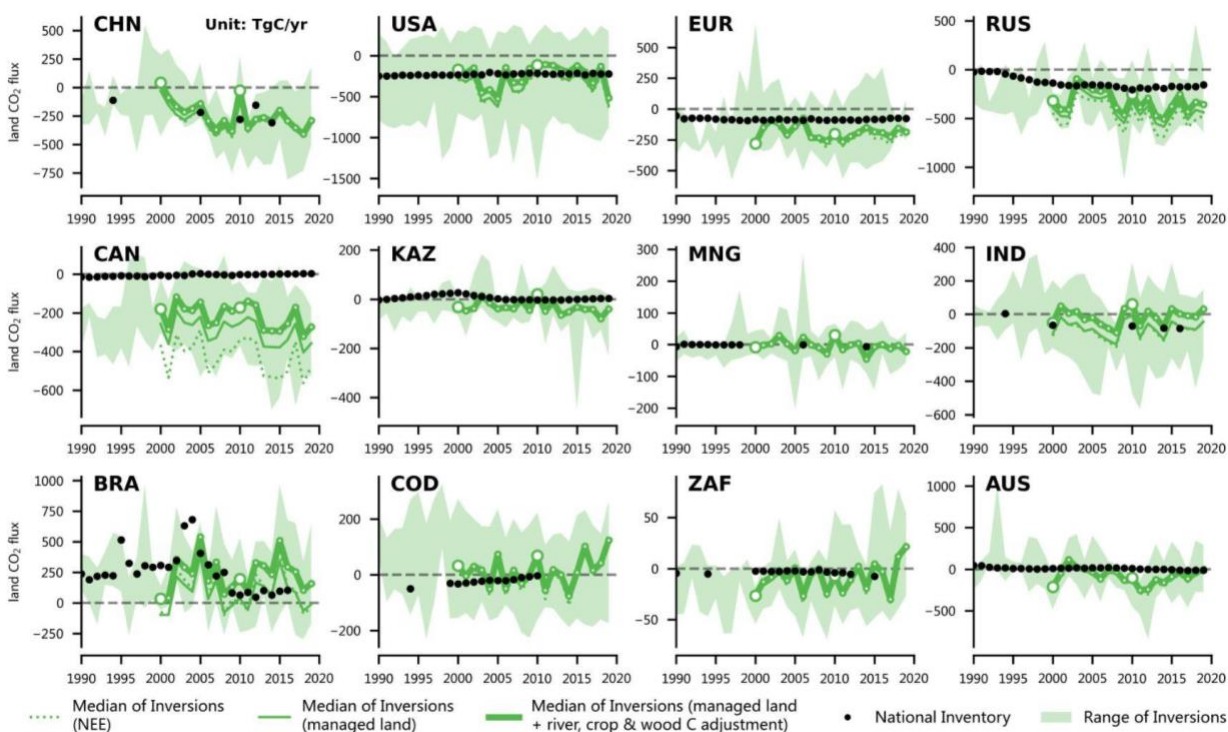

**Figure 3. Land-atmosphere $CO_2$ fluxes (TgC yr$^{-1}$) from China (CHN), United States (USA), EU27 & UK (EUR), Russia (RUS), Canada (CAN), Kazakhstan (KAZ), Mongolia (MNG), India (IND), Brazil (BRA), Democratic Republic of the**

**Congo (COD), South Africa (ZAF), and Australia (AUS). By convention, $CO_2$ removals from the atmosphere are counted negatively, while $CO_2$ emissions are counted positively. The black dots denote the reported values of the sum**





**of the land use, land use change and forest (LULUCF), agriculture and waste sectors from national inventories. Note that C stock change for agricultural land is reported under LULUCF whereas CH$_4$ and N$_2$O emissions of agricultural activities are reported in the agriculture sector. In the agricultural sector, fossil emissions CO$_2$ from agricultural**

**machinery were excluded. For the EUR, the NISMON inversions data were removed in 2018 and 2019, being a large outlier. The green solid thick lines denote the median of land fluxes over managed land of all CO$_2$ inversions, after adjustment of CO$_2$ fluxes from lateral transport by rivers, crop and wood trade. The solid thin line is the median of inversions over managed land and without lateral transport adjustment. The dotted line is the original median of inversions, where the large hollow dots in the line are for years beginning a decade and small hollow dots for other**

**years. Light green shading from the min-max range of inversions. Since before 2000, there are only 4 inversion models, the median is not shown.**

In large fossil CO$_2$ emitter countries of temperate latitudes, inversions and NI estimates are pretty similar in China (CHN) and USA, but give a higher CO$_2$ uptake in EU27 & UK (EUR), Russia (RUS), and Canada (CAN) (Fig. 3). In these five countries,

correcting inversions by CO$_2$ fluxes induced by river carbon transport and by the trade of crop and wood products tends to lower CO2 sinks, especially for large crop exporters like USA and CAN. But it still leaves a median CO$_2$ uptake after adjustment (of 258 TgC yr$^{-1}$ in CHN, 247 TgC yr$^{-1}$ in USA, 185 TgC yr$^{-1}$ in , 326 TgC yr$^{-1}$ in RUS, and 219 TgC yr$^{-1}$ in CAN during 2001-2019), which are still higher than NI reports (Fig. 3). The misfit of NI with inversions differs between countries, however.


In CHN, the six successive national communication estimates fall in the range of the six inversions, and give a trend towards an increasing carbon storage. Corrected inversions provide a median CO$_2$ sink of 142 TgC yr$^{-1}$ in 2005 and of 245 TgC yr$^{-1}$ during 2010-2014, consistent with reported values from inventory reports (166 TgC yr$^{-1}$ in 2005, and an average of 247 TgC yr$^{-1}$ in 2010, 2012 and 2014). China's fourth national communication (also the first BUR) in 2012 reported sinks from forest

and woody biomass, but other communications counted grasslands, wetlands, etc which may explain the smaller sink of the NI in 2012.

In the USA, the carbon stock change estimates of the NI fall within the range of inversions during the last three decades, with a mean value of 221 TgC yr$^{-1}$ in the NI during 2000-2019 compared to 243 TgC yr$^{-1}$ by inversions. Yet, the USA inventory

gives a small decrease of carbon sinks with time, whereas corrected inversions produce a strong decrease of the net CO$_2$ uptake, from an average of 287 TgC yr$^{-1}$ in the 2000s to 200 TgC yr$^{-1}$ in the 2010s, dropping by near 30% during the last 30 years despite of the uncertainty suggested by the range of inversion model results. Estimates from NIs also show a decreasing trend but with less fluctuation from a mean value of 239, 222, 219 TgC yr$^{-1}$ in the 1990s, 2020s, 2010s respectively. In EUR, inversions systematically indicate a larger net CO$_2$ uptake than NI, by on average around 104 TgC yr$^{-1}$ larger than NI, yet with

a non significant  trend (Mann Kendall test p = 0.7) consistent with stable land carbon storage shown by inventories.





In the two largest boreal and arctic countries Canada (CAN) and Russian (RUS), inversions produce a $CO_2$ sink (average 219 and 326 TgC yr$^{-1}$) which is systematically larger than the NIs (2 and 173 TgC yr$^{-1}$, respectively) during 2001-2019. The larger Russian sink of inversions is similar with the results of a recent analysis (Schepaschenko et al., 2021)of forest inventory data estimating a carbon accumulation of 343 TgC yr$^{-1}$ from 1988 to 2014. The Russian carbon stock increase is 6.0 TgC yr$^{-1}$ annually in the NI during the 2000s, smaller than the increasing $CO_2$ sink rate of 16.4 TgC yr$^{-1}$ across inversions. When inversions include all lands in these two countries instead of managed land only, the net land $CO_2$ sink becomes 40% larger in CAN and 16% larger in RUS. Unmanaged lands cover 30% of CAN and 15% of RUS in our intact forest dataset, respectively, associated with nearly identical $CO_2$ sink densities of 44 gC m$^{-2}$ yr$^{-1}$ and 29 gC m$^{-2}$ yr$^{-1}$, compared to 46 gC m$^{-2}$ yr$^{-1}$ and 30 gC m$^{-2}$ yr$^{-1}$ in managed lands. It should be noted that in Canada's NIR, flux from grassland is not estimated in its national inventories, but the area of intensively managed grassland only accounts for 1% of total area.

Among the selected large forested tropical countries, Brazil (BRA) is one of the few non-Annex I countries that provided continuous time series of NIs since 1990. The Brazilian NI shows a net loss of carbon stocks from 1990 to 2020, with an increasing loss from 1990 to 2005, followed by a decrease afterwards. This change is explained mostly by deforestation rates (tree cover loss) which declined by a quarter between 2001-2005 (3.4 Mha yr$^{-1}$) and 2006-2016 (2.7 Mha yr$^{-1}$) (Global Forest Watch, 2021), following strict government policies to protect forests, which were enforced until 2019. The latest year of reported inventory was 2016 in BRA, but satellite estimates of deforestation rates in Brazilian Amazon from the Program to Calculate Deforestation in the Amazon (PRODES) system of the Brazilian Space agency indicated a sharp increase of deforestation in 2019 (10,129 km$^2$), with a forest area loss 25% higher than the average of 2006-2016 (8,119 km$^2$ yr$^{-1}$) (Silva Junior et al., 2020). On top of deforestation, degradation and fires also cause a loss of carbon in BRA and other Amazon countries, such as Peru (26 TgC yr$^{-1}$ during 2001-2019) and Colombia (18 TgC yr$^{-1}$ during 2001-2019). The area of degraded forests has been reduced in BRA tailing off with the reduction trend of deforestation until 2019 (Matricardi et al., 2020; Bullock et al., 2020), even though the components of degradation from burned and logged forests, two processes causing the largest loss of carbon per unit area, have remained constant over time. The number of active fires in BRA seems to have stayed constant, with peaks during dry years, with $CO_2$ emissions from fires that may be larger and decoupled from decreasing $CO_2$ losses from decreasing deforestation (Aragão et al., 2018). Drought generally causes abnormal $CO_2$ losses in the Brazilian Amazon, of 0.48 GtC yr$^{-1}$ during the 2010 drought, based on a regional inversion with aircraft $CO_2$ vertical profiles (Gatti et al., 2014) and of 0.25 GtC yr$^{-1}$ during the extreme El Niño drought of 2015, from above ground biomass loss estimated by satellite vegetation optical depth changes (Qin et al., 2021). The land fluxes of inversions indicates Brazilian land became abnormal sources in dry years 2005 (540 TgC yr$^{-1}$), 2007 (334 TgC yr$^{-1}$), 2010 (195 TgC yr$^{-1}$) and 2015 (511 TgC yr$^{-1}$), with a sudden net increase compared to the previous year (240 TgC in 2004, 180 TgC in 2006, 132 TgC in 2009 and 232 TgC in 2015). Over the period 2010-2019, the above ground net mean $CO_2$ flux of the Brazilian Amazon area was estimated to be a weak source of 0.06 GtC yr$^{-1}$ (Qin et al., 2021), also consistent with data from the inversion of Palmer et al. (2019) (see their



Table 1). The median land $CO_2$ flux of inversions over the same period show a source of 0.25 GtC yr$^{-1}$, comparable in magnitude, but with a large spread. Recently, a top-down estimate based on 2010-2018 aircraft profiling of CO2 mole fractions (Gatti et al. 2021) suggested a substantial source of carbon in the eastern Amazon forest basin driven by fire emissions and loss of forest carbon uptake in dry seasons. The western part of the basin was near-neutral in NEE, with deforestation fires and climate warming/drying playing a much smaller role. We also acknowledge that the estimate by Qin et al. (2021) gives only

the carbon change in above-ground biomass, not strictly comparable to inversion results, the later including soil and inland water $CO_2$ fluxes, and legacy $CO_2$ emissions following mortality from the decomposition of coarse woody debris (Yang et al., 2021). Note also the importance of lateral carbon fluxes from export of agricultural commodities in Brazil, as a driver of deforestation (Follador et al. 2021; Weisse and Goldman 2021).As for the another selected large forested tropical country, the Democratic Republic of Congo (COD), inventories show a net sink of 19 TgC yr-1 with a smaller interannual variability than

in BRA despite a similar forested area. Interestingly, NIs in COD show a decreasing $CO_2$ sink from 1994 to 2010, while inversions give an increasing $CO_2$ sink from the 1990s to the period around 2010, followed by a reversal after 2010. During the last decade, years after 2015 were net $CO_2$ sources to the atmosphere for COD. It should be acknowledged that the NI of COD is extremely uncertain (and contradictory information has been provided by the country in different official documents).

For South-East Asian maritime continent countries, that is, Indonesia (IDN), Malaysia (MYS), Papua New Guinea (PGN) grouped together (Fig S4), we found a large peak of $CO_2$ emissions during the El Niño of 1998 corresponding to extreme fire emissions from peat burning (Page et al., 2002). This group of countries shows a net sink of 60 TgC yr$^{-1}$ of $CO_2$ since 2000. For continental Southeast Asian countries, that is, Thailand (THA), Myanmar (MMR), Laos (LAO), Cambodia (KHM) and Viet Nam (VMN) grouped together, we found on average that inversions give a similar net $CO_2$ flux as reported by NIs. In

this group of countries, inversions give a decreasing sink trend in the last decade (Fig S4), consistent with the observation of increased forest clearing and biomass carbon losses, in particular over mountain regions (Zeng et al., 2018; Davis et al., 2015).

For India (IND), although the land $CO_2$ sinks show an increased uptake across inversions during the first half of the 2000s with an annual uptake of 14 TgC yr$^{-1}$ during this period, the $CO_2$ uptake from the inversions fluctuated between positive and negative values in the 1990s and 2010s, indicating that the role of land $CO_2$ flux shifted between a net carbon sink and a net

carbon source (Takaya et al., 2021). This shift to a net $CO_2$ source could be explained by a decreased Indian monsoon after ~2007 (Univ Hawaii JJA Indian Monsoon Index, http://apdrc.soest.hawaii.edu/projects/monsoon/seasonal-monidx.html, last access: 05 July 2021). For the two temperate continental countries, Mongolia (MNG) and Kazakhstan (KAZ), the land $CO_2$ flux fluctuates around zero with a small interannual variation, indicating a stable trend of land flux changes and a small

contribution to the uptake of all northern hemisphere Annex I countries. For Australia (AUS), there is a clear $CO_2$ sink anomaly during the extremely wet La Niña event from May 2010 to March 2012 (Haverd et al., 2016; Poulter et al., 2014). In the following fire season of late 2012 / early 2013, more fires were reported from the legacy of a higher fuel load in the previous wet period, and these $CO_2$ emissions likely caused the net $CO_2$ uptake to have decreased (Harris and Lucas, 2019).



**3 Results for CH₄ anthropogenic emissions**

**3.1 Total anthropogenic CH₄ emissions**

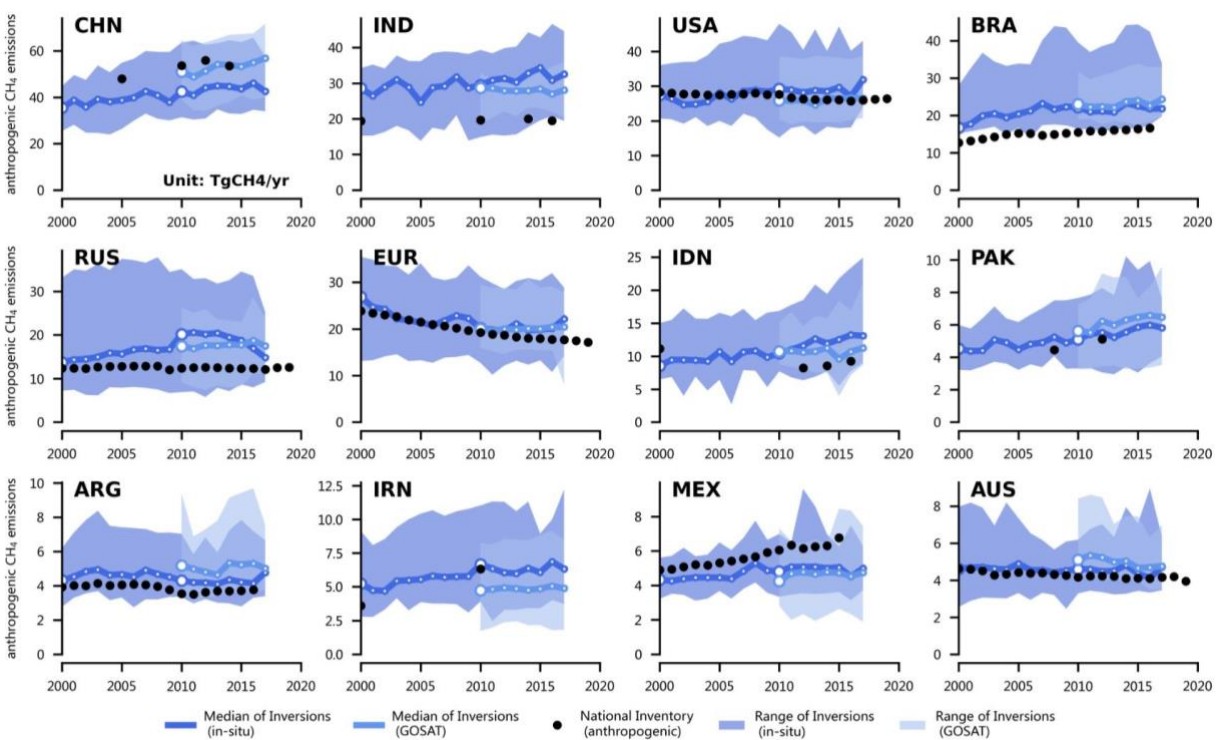

**Figure 4. Total anthropogenic CH₄ fluxes for the 12 top emitters: China (CHN), India (IND), United States (USA), Brazil (BRA), Russia (RUS), EU27 & UK (EUR), Indonesia (IDN), Pakistan (PAK), Argentina (ARG), Iran (IRN),**
**Mexico (MEX), and Australia (AUS), following Method 1 based on the original separation of anthropogenic vs. natural sources by each inversion with wildfires subtracted (see section 1). The black dots denote the reported values from national inventories. The dark blue lines/areas denote the median and maximum-minimum ranges of in-situ CH₄ inversions and the light blue ones of GOSAT inversions, respectively. Note the different scale for each country.**

Figure 4 shows the variations of CH₄ anthropogenic emissions from 2000 to 2019 (2017 for inversions), defined by summing the sectors of agriculture and waste, fossil fuels, and biomass and biofuel burning for the 12 selected countries, assuming that all biomass burning is anthropogenic (see section 1.4). The distribution of emissions is strongly skewed even among the top 12 emitters, with the largest and most populated countries forming a group of super-emitters and other countries having much smaller emissions, thus more difficult to quantify by inversions. According to the GOSAT inversions, China (CHN) has the





largest emission of around 53 TgCH$_4$ yr$^{-1}$, followed by India (IND) with 28 TgCH$_4$ yr$^{-1}$, the USA with 26 TgCH$_4$ yr$^{-1}$, Brazil (BRA) with 23 TgCH$_4$ yr$^{-1}$, EU27 & UK (EUR) with 20 TgCH$_4$ yr$^{-1}$, Russia (RUS) with 18 TgCH$_4$ yr$^{-1}$ and Indonesia (IDN) with 11 TgCH$_4$ yr$^{-1}$, and the other countries of only around 5 TgCH$_4$ yr$^{-1}$. Note the asymmetric range around the median of inversions for BRA in Fig 4. The data in Fig 4 indicate a large spread between inversions, owing to differences in model settings and transport. Differences due to different methods used to separate anthropogenic from natural emissions are smaller

than this spread and we discuss them in section 5.2. We observed on average a smaller range of GOSAT inversions (number of inversions: n = 11) than in-situ inversions (n =10) in countries emitting more than 10 TgCH$_4$ yr$^{-1}$. The median emissions from GOSAT inversions are systematically lower than from in situ, except in CHN where GOSAT inversions are on average 21% larger during 2010-2017. Ranges overlap between the two inversion ensembles. Generally, the difference between NIs and inversions is of the same sign based on GOSAT and in-situ, which gives us some confidence for evaluating NIs because

the GOSAT observations are different and independent from in-situ networks. Ex-ante, we trust more GOSAT based inversions over most countries, because GOSAT has a better observation coverage, except for EUR and USA where the in-situ network is dense (Fig S2). From the result shown in Fig 4 that GOSAT and in-situ inversions are on the same side compared to NIs, we can thus be more confident of results provided by in-situ inversions over the full time period.

For China (CHN), during 2000-2017, the median of anthropogenic emissions from in situ inversions is 44 TgCH$_4$ yr$^{-1}$ lower than all the NI reports (53 TgCH$_4$ yr$^{-1}$, average of 4 observations in 2005, 2010, 2012 and 2014), but the median of GOSAT inversions (53 TgCH$_4$ yr$^{-1}$) is close to NIs (54 TgCH$_4$ yr$^{-1}$) in the 2010s (Fig 4). The trend of emissions is consistent between inversions and NI data, although the increasing trend is larger in GOSAT than in-situ after 2010. For the USA, the median of inversions is close to the NI reported value during the whole study period. The trend of in-situ inversions in the USA is positive,

with an increase of 0.3 TgCH$_4$ yr$^{-1}$ from 2000 to 2017, opposite to the small decrease of 0.1 TgCH$_4$ yr$^{-1}$ in the NI data. The GOSAT inversions also show a positive trend of 0.1 TgCH$_4$ yr$^{-1}$ from 2010 to 2017. This different trend between inversions and the US national inventory might be attributed to CH$_4$ leakage from unconventional oil and gas extraction not fully accounted for in NI (Allen, 2016). This type of oil and gas production became important after the mid 2000s and has emission factors double the values from the USEPA currently used in the NI, as shown by local and regional measurements campaigns

(Alvarez et al., 2018).

For the EU27 & UK (EUR), the decreasing trend of anthropogenic CH$_4$ emissions diagnosed by inversions is consistent with NI estimates, but the median of inversions is higher than the NI data by 2 TgCH$_4$ yr$^{-1}$ for the study period from 2010 to 2017. This result is consistent with the EU synthesis results of Petrescu et al. (2021a) although they used a different method to subtract natural emissions, based on regional estimates of peatlands and inland water natural emissions. Positive differences

between inversions and NI reports are found for Russia (RUS), India (IND), Brazil (BRA), Argentina (ARG) and Australia (AUS). In Russia (RUS), emissions are larger in inversions than in the NI data, and this result is robust to the choice of the method to separate natural from anthropogenic sources (see Fig S5 and Fig 10). Note however that RUS wetlands emissions are partly concentrated in the region of Western Siberia (northern Ob river basin) just to the south of a major basin of gas




extraction, and these two sources are difficult to separate from each other in global inversions. Note that methane emissions

from fires in Russia are smaller than fossil and wetland emissions, and fires cannot explain why inversions give larger emissions than the NI. In Brazil (BRA), the result that inversions give systematically larger $CH_4$ emissions than NI is also robust due to the method used to separate wetlands from anthropogenic emissions (Fig S5), but our inversions did not use in-situ data in the interior of the country (Fig S1) and the coverage of GOSAT is sparse due to clouds (Fig S2), especially in the wet season, which makes the estimation of the total $CH_4$ source uncertain over this country. In India (IND), both the in situ

and GOSAT inversions also give a higher anthropogenic emission than the NI data and the share of emissions from natural wetlands is much smaller than in RUS and BRA, reducing the risk of aliasing anthropogenic for natural emissions, and suggesting an under-reporting by the NI. In Indonesia (IDN) the median of inversions is slightly higher than the NI data with the separation method used in Fig 4, but close to NI with other methods (Fig 10 and Fig S5). All inversions give a large positive anomaly of $CH_4$ emissions during the 2015 El Niño, when abnormal peat fire emissions occurred (Heymann et al., 2017; Yin

et al., 2016), a biomass burning event being attributed to an anthropogenic source by our aggregation of inversion results (section 1) but likely not included by the NI. In AUS, the anthropogenic emissions mainly from the coal extraction and cattle sector of the NI are found to be very close to the inversion median, both across in situ and GOSAT inversions. In Pakistan (PAK) and Iran (IRN), NI values show good consistent with the in-situ inversions, however, in Pakistan (PAK) the GOSAT inversions are 18% higher than the in-situ inversions. Mexico (MEX) is the only country whose NIs are higher than the

inversions among the 12 selected countries, which is mainly attributed to the difference between inversions and inventories in the agriculture and waste sector (see below).

## 3.2 Fossil CH₄ emissions

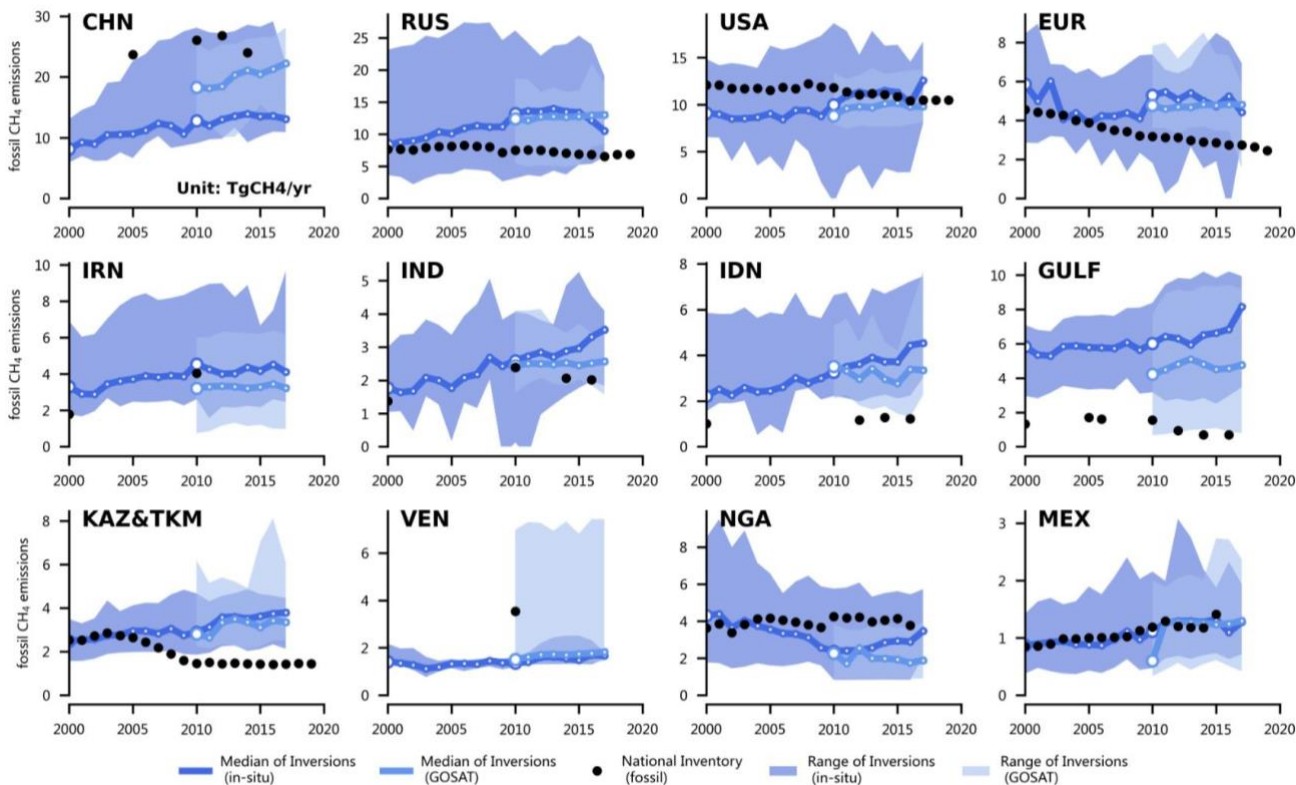

**Figure 5. CH₄ emissions from the fossil fuel sector from the top 12 emitters of this sector: China (CHN), Russia (RUS), United States (USA), EU27 & UK (EU), Iran (IRN), India (IND), Indonesia (IDN), Persian Gulf countries (GULF = Saudi Arabia + Iraq + Kuwait + Oman + United Arab Emirates + Bahrain + Qatar), Kazakhstan & Turkmenistan (KAZ&TKM), Venezuela (VEN), Nigeria (NGA), and Mexico (MEX). The black dots denote the reported value from national inventories. In the NI data shown in Fig 5 for GULF, Saudi Arabia reported four NIs in 1990, 2000, 2010, and 2012, Iraq reported one in 1997, Kuwait reported three in 1994, 2000, and 2016, Oman reported one in 1994, United Arab Emirates reported four in 1994, 2000, 2005 and 2014, Bahrain reported three in 1994, 2000 and 2006, and Qatar reported one in 2007. The reported values are interpolated over the study period to be summed up and plotted in the figure. For KAZ&TKM, the reported values of Turkmenistan during 2001-2003, 2005-2009, 2011-2020 are interpolated and added to annual reports from Kazakhstan, an Annex I country for which annual data are available. Other lines, colors and symbols as Fig 4.**

Fig 5 presents the CH₄ emissions for the top 12 emitters from the fossil sector. The largest emitter is China (CHN), from the sub-sector of coal extraction mainly (85% in 2014), followed by Russia (RUS) and the United States (USA). The range of inversions relative to median values is larger for fossil emissions than for total anthropogenic emissions, reflecting the fact



that the uncertainty of inversions increases through their separation of fossil from other sources. Here, GOSAT inversions in
which fossil sector emissions were separated from the total emission in each grid cell using the share provided by a prior, differ
from in situ inversions where different sectors correspond to specific tracers, in particular for CHN where the choice of a prior
to separate fossil from other emissions is critical (Liu et al., 2021). In China (CHN), both in-situ and GOSAT inversions find
on average significantly smaller emissions than the NI, by 50% (13 $TgCH_4$ $yr^{-1}$) for in-situ and 24% (6 $TgCH_4$ $yr^{-1}$) for GOSAT

in the 2010s, consistent with Liu et al. (2021). In Russia (RUS), in-situ and GOSAT inversions both have larger fossil emissions
of near 6 $TgCH_4$ $yr^{-1}$ than NI, with a diverging trend of an increase in inversions and a decrease in the NI. This mismatch is
possibly due to aliasing between wetland emissions and gas extraction industries that occur in roughly the same region, or
because of accidental leaks from ultra-emitters that are ignored in the NI. The ultra-emitters defined by Lauvaux et al. (2021)
are namely all short duration leaks from oil and gas facilities (e.g. wells, compressors) with an individual emission > 20 $tCH_4$

$h^{-1}$, each event lasting generally less than one day. The contribution of these ultra-emitters is discussed in section 3.3.1. In the
USA, fossil emissions from in-situ inversions are smaller than in NI by 26% until 2011 and then aligned. The gap between NI
and inversions may be compensated by: 1) under-reported emissions (Alvarez et al.) in inventories from the shale gas extraction
industry that are representing today 68% of the total USA oil and gas production (EIA, 2021b), and 2) excluded oil and gas
offshore emissions in our all fossil sources top-down budget through the land masking applied to the inversions. We note that

although the emissions from the offshore sector might be underestimated (Gorchov Negron et al., 2020), it produces only about
3% of the total U.S. natural gas production (EIA, 2021c). In the EUR, a similar fossil emission is found from in-situ inversions
and NIs before 2010, but both in-situ and GOSAT inversions show higher emissions to NI from 2010 to 2017. The decreasing
trend of fossil emissions between 2000 and 2010 is very consistent between inversions and NI reports in EUR. In India (IND),
fossil emissions mainly come from fugitive emissions (~60% from natural gas and 30% from solid fuels in 2010 from MoEFCC

(2015)). Only three years are available from NI, and they report similar values than GOSAT inversions with constant emissions
from 2010. On the contrary, in-situ inversions suggest continuous increasing emissions.

In major oil and gas extracting countries that have negligible agricultural and wetland emissions like Kazakhstan (KAZ), Iran
(IRN), Persian Gulf countries (GULF), fossil emissions should be easier to separate by inversions and thus to be compared

with NI. We found that GULF and KT emissions are three and two times higher as diagnosed by in-situ and GOSAT inversions
compared to NI reports, respectively. The reasons for the GULF apparent under-reporting could be because of ultra-emitters
not included in the NI, a point further examined in section 5. Note that SAU emissions seem to be lower than other GULF
countries according to inversions, but SAU is not separated well by inversions from neighbouring countries. More ultra-
emitters and larger emissions budgets from ultra-emitters (see section 5) were also found in Qatar, Kuwait, Iraq than in SAU

(Lauvaux et al., 2021). Similarly, KAZ is downwind of Turkmenistan (TKM), which has a high share of ultra-emitters
(Lauvaux et al., 2021), and global inversions working at rather coarse resolution could mis-allocate to KAZ emissions coming
from TKM. The emitting countries in the Persian Gulf area have no atmospheric $CH_4$ in-situ station coverage, while KAZ has
two stations. In contrast, the sampling of atmospheric column $CH_4$ by GOSAT is rather dense in all those countries, thanks to



frequent cloud free conditions. Thus GOSAT inversions could be viewed as more accurate than in-situ inversions for IRN,
SAU, KAZ and we note that GOSAT inversions give lower emissions than in-situ inversions. We also compared inversions
and NIs with annual $CH_4$ emissions data compiled by PRIMAP-HIST (Gütschow et al., 2016) for the Energy sector and found
that this dataset produces much larger emissions than NIs and the median of inversions for GULF and KAZ&TKM (Fig S6).
The methodology of PRIMAP-HIST interpolates and extrapolates UNFCCC values using trends of EDGAR V4.2, an inventory
which is known to overestimate fossil $CH_4$ emissions (Thompson et al., 2015; Patra et al., 2016; Ganesan et al., 2017). The
higher values of PRIMAP-HIST may thus be due to the extrapolation of temporally sparse national inventories for those
countries, and this dataset should not be considered as similar to NIs for the fossil fuel $CH_4$ sector.

In Nigeria (NGA) and Venezuela (VEZ), where nearly half of the oil and gas industry is offshore or near the coast (NAPIMS,
2021), we found that fossil $CH_4$ emissions are smaller in inversions than in the NIs. This result should be considered with
caution as those countries have a small size, thus with emissions difficult to constrain by global inversions, the presence of
clouds that reduces the number of GOSAT soundings, and anthropogenic and natural $CH_4$ emissions collocated with fossil
ones. Finally, for Mexico (MEX), GOSAT and in-situ inversions show good agreement with respect to NI. However, this
apparent agreement might stem from both an overestimation of offshore emissions (not included here in inversions due to land
masking ) and an underestimation of inland fossil fuel emissions by the NI (Zavala-Araiza et al., 2021). Possible reasons could
include: 1) In MEX, roughly 80% of oil production and 60% of gas production is from offshore shallow water wells. Emission
inventories seem to be overestimating offshore emissions of $CH_4$ by about an order of magnitude (Zavala-Araiza et al., 2021);
2) Emission factors used in bottom-up inventories are generic (not specific for Mexican type of wells, reservoir, technology,
age of technology, etc.); 3) Due to the elongated shape of Mexico and because it is surrounded by water, the spread of the
inversions are higher compared to other countries.


## 3.3 Agriculture and waste CH₄ emissions

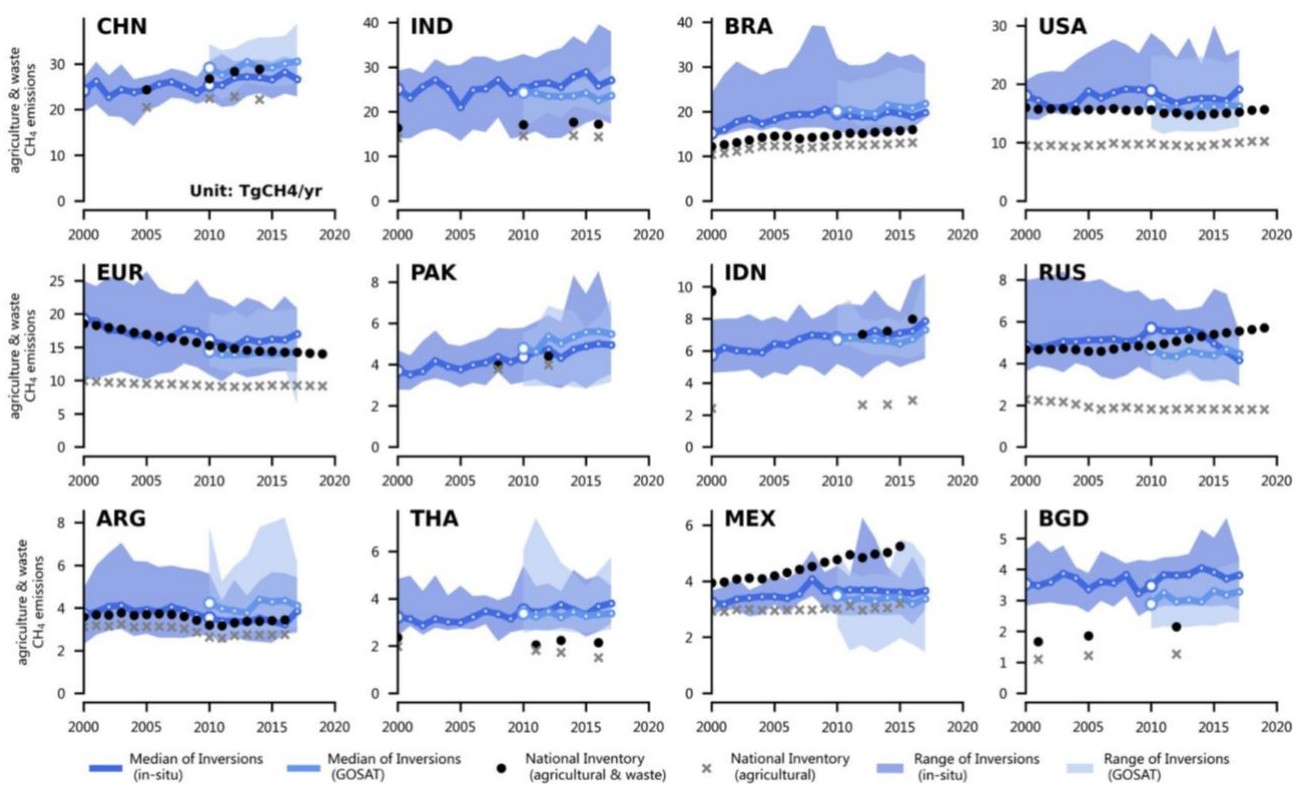

**Figure 6. CH₄ emissions from agriculture and waste for the 12 largest emitters in this sector, China (CHN), India (IND), Brazil (BRA), United States (USA), EU27 & UK (EUR), Pakistan (PAK), Indonesia (IDN), Russia (RUS), Argentina (ARG), Thailand (THA), Mexico (MEX), and Bangladesh (BGD). The black dots denote the reported estimates from NIs from the agriculture and waste sector and the grey crosses denote NIs emissions from the agriculture sector only ; the difference being the waste sector. Other lines, colors and symbols as Fig 4.**

Fig 6 presents the CH₄ emissions of the agriculture and waste sector for the top 12 emitters. Like in Fig 5, the relative spread of inversions (range divided by mean) is larger for this sector than for the total of all anthropogenic sectors. We observed that the median of GOSAT inversions is close to the median of in situ inversions within ± 0.3 TgCH₄ yr⁻¹ over the period 2009-2017 across the countries in Fig 6. The values from NIs also show good consistency with the inversions. In China (CHN), agriculture and waste emissions from the most recent NI reports in the 2010s (28 TgCH₄ yr⁻¹) are between the GOSAT inversions (29 TgCH₄ yr⁻¹) and in-situ inversions (27 TgCH₄ yr⁻¹). The trend in agricultural emissions is consistent between inversions and NI for CHN. In India (IND), inversions give systematically higher emissions than NI by 50% for GOSAT and 38% for in-situ, with GOSAT and in-situ inversions being similar in 2010 (~24 TgCH₄ yr⁻¹), and showing thereafter a decreasing trend in GOSAT (-0.1 TgCH4 yr-1) compared to an increasing trend in in-situ (+0.3 TgCH₄ yr⁻¹). In IND, from the



national inventory, enteric fermentation is the major $CH_4$ source of the Agriculture and Waste sector, contributing 61% of emissions while 20% for rice cultivation and 16% for waste. A similar result is found in Bangladesh (BGD), where agricultural

emissions are dominated by rice production (48% in 2012) and enteric fermentation (42% in 2012), with GOSAT inversions nearly double than the NI reports during 2001 and 2012. The large differences between the inversions and NI for IND and BGD could be due to the potential underestimation of livestock $CH_4$ emissions by NI. NI used the Tier 1 method and associated emission factors from the 2006 IPCC Guidelines for National Greenhouse Gas Inventories (Eggleston et al., 2006), while a recent study (Chang et al., 2021b) found that the estimates using the revised Tier 1 or Tier 2 methods from the 2019 Refinement

to the 2006 IPCC Guidelines for National Greenhouse Gas Inventories (Buendia et al., 2019) are 48%-60% and 42%-61% higher for IND and BGD by 2010, respectively, and match better the inferred emissions from inversions. In Brazil (BRA), both GOSAT and in-situ inversions are systematically larger than the NIs by 34% and by 29% respectively, but show consistent increasing trend over their study periods. In the USA, the medians of GOSAT and in-situ inversions are slightly higher than the NIs, while NIs show a slow decreasing trend over the study period. In Indonesia (IDN), Pakistan (PAK) and Argentina

(ARG), the medians of in-situ inversions have a good consistency with NI reported values, while GOSAT inversion emissions in the 2010s are on average 19% higher in Pakistan, 24% higher in Argentina but 9% lower in Indonesia compared to the NI reports. In the EU27 & UK (EUR), emissions from agriculture and waste are found to have significantly decreased over time in the NI data, mainly from solid waste disposal (Petrescu et al., 2021b), a trend that is captured by inversions and close to the NIs over the study period. In contrast, emissions from agriculture and waste in Russia (RUS) are reported to have a positive

trend after 2010, contributed mainly from the solid waste disposal (crosses vs. circles in Fig. 6), whereas in-situ inversions produce a consistent trend from 2000 to 2014 but a decrease thereafter, and the GOSAT inversions produce stable values, lower than the NI after 2010. Last, in Mexico (MEX), the inversion data in Fig 6 indicate a consistently lower agricultural and waste emission than the NI, by 1.6 $TgCH_4$ $yr^{-1}$ across in-situ inversions, and by 1.0 $TgCH_4$ $yr^{-1}$ in the GOSAT inversions. Inversions produce stable emissions in the period after 2010, whereas the NI gives an increase at a rate of 2% yr-1, mainly

from the solid waste disposal (~60%) and livestock (~40%). Note that livestock $CH_4$ emissions in Mexico increased by more than 20% during 2000-2018 (from ~2.0 to ~2.4 $TgCH_4$ $yr^{-1}$) from all methodologies used by Chang et al. (2021b) (2006 Tier 1, 2019 Tier 1 and Mixed Tier) suggesting that the increase of the national inventory agricultural emissions shown in Fig 6 is consistent with more recent methodologies.



## 785  4 Results for N₂O emissions

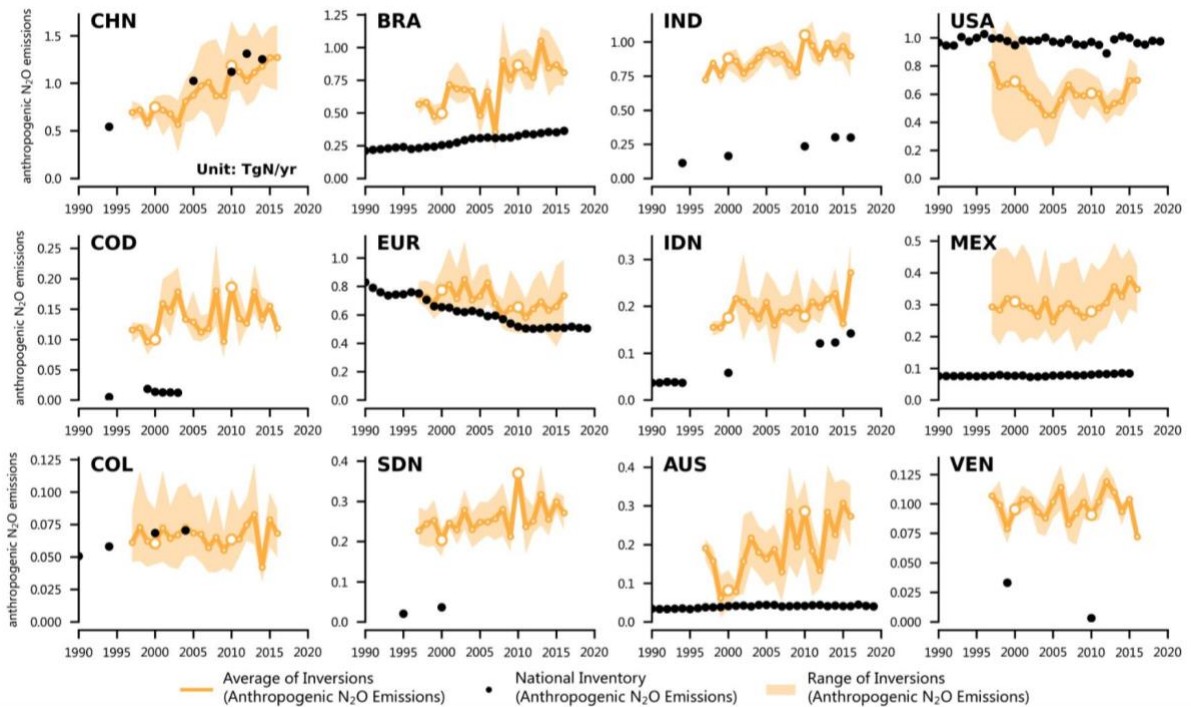

**Figure 7. Anthropogenic N₂O fluxes of the top 12 emitters: China (CHN), Brazil (BRA), India (IND), United States (USA), Democratic Republic of the Congo (COD), EU27 & UK (EUR), Indonesia (IDN), Mexico (MEX), Colombia (COL), Sudan (SDN), Australia (AUS), and Venezuela (VEN). The black dots denote the anthropogenic emissions from**
**the UNFCCC national inventories. The thick orange lines denote the median of anthropogenic fluxes among N₂O inversions (thick) and the light orange areas denote the maximum-minimum ranges of all inversions.**

Figure 7 compares anthropogenic N₂O emissions from inversions and inventories. Inversions tend to produce higher emissions than inventories, except for the USA, China and Colombia. In all the countries, inversions also show a larger interannual

variability than NI data. In the USA, the median of anthropogenic emissions from inversions is about 30% lower than the inventories, and shows a larger interannual variability with a minimum around the year 2005. In the EU27 & UK (EUR), the median of inversions became 32% larger than NIs after 2013, but inversions capture the decreasing trend of reported emissions before that year. This decreasing trend was attributed mainly to industrial emissions, according to NI data and other inventories analyzed by Petrescu et al. (2021a). In general, the masking of unmanaged lands in gridded inversion fluxes reduces national

emissions, in particular in tropical countries like Democratic Republic of the Congo (COD) and Brazil (BRA) where unmanaged forests are significant emitters of N₂O (see Fig 12). Possible reasons for underestimated anthropogenic emissions for nearly all the non-Annex 1 countries can be the use of Tier 1 emission factors (EF) which may be lower than when soil and





climate dependence is accounted for (Philibert et al., 2013; Shcherbak et al., 2014; Wang et al., 2020b), and the non-linear observed concave response of cropland soils emissions as a function of added N fertilizers (Zhou et al., 2015) which makes

emissions higher than the linear relation used by NIs in Tier 1 approaches. In an ideal world, the EF should represent the natural and anthropogenic components since these cannot be distinguished from field measurements, from which EFs are derived. In practice, the EFs are mostly based on measurements made in temperate climates and for soils of cropland established long ago with little 'background' emissions, so there may be a systematic underestimation of default IPCC EFs of emissions from tropical climates and recently established agricultural land. Another reason may be the omission of emissions

from reactive N contained in organic fertilizers (manure), about which NI do not provide details for non-Annex I reports. Last, anthropogenic indirect emissions (AIE) from atmospheric nitrogen deposition and leaching of human induced nitrogen additions to aquifers and inland waters are reported by Annex I countries using simple emission factors, but they are not systematically reported by non-Annex I countries. In Table S2, we compiled AIE data for the 20 largest $N_2O$ emitting countries in the non-Annex I category, including the ones displayed in Fig 7. Those indirect emissions represent 18% of direct mineral

fertilizer induced anthropogenic emissions from cropland soils in EUR and 16% in the USA for instance, and thus make NI data systematically lower than inversions for countries that did not include them. According to the data in Table S2, indirect emissions represent 5 to 10% of anthropogenic emissions in most of the non-Annex I countries shown in Fig. 7. In consequence, their omission cannot explain all the mismatch between NIs and inversions. We also compared in Table S2 indirect emissions data from inventories with those provided by the FAOSTAT database (FAO, 2021).

**5 Discussion**

In this section, we further analyse the three gases comparison between NI and inversions. First, we compare the land $CO_2$ flux to fossil fuel emissions and their respective trends. Then we discuss the uncertainty arising from the separation between anthropogenic and natural $CH_4$ emissions in inversions by comparing the results of different separation methods, and we analyze how inversions resolve fossil versus agricultural emissions budgets in each country. The contribution of $CH_4$ emissions

from ultra-emitters in the fossil fuel sector, which is not counted in inventories but could explain why inversions diagnose higher emissions than inventories in many oil and gas extracting countries, is further analyzed using independent estimates. Then, we analyse the separation of natural from anthropogenic sources in national $N_2O$ budgets from inversions, a topic which has not been addressed in previous studies. Finally, we compare the results of our global inversions with regional inversions based on higher resolution transport models or assimilating regional data for China (CHN), EU27 & UK (EUR), USA and

Brazil (BRA).

**5.1 Land $CO_2$ fluxes compared to fossil fuel emissions**





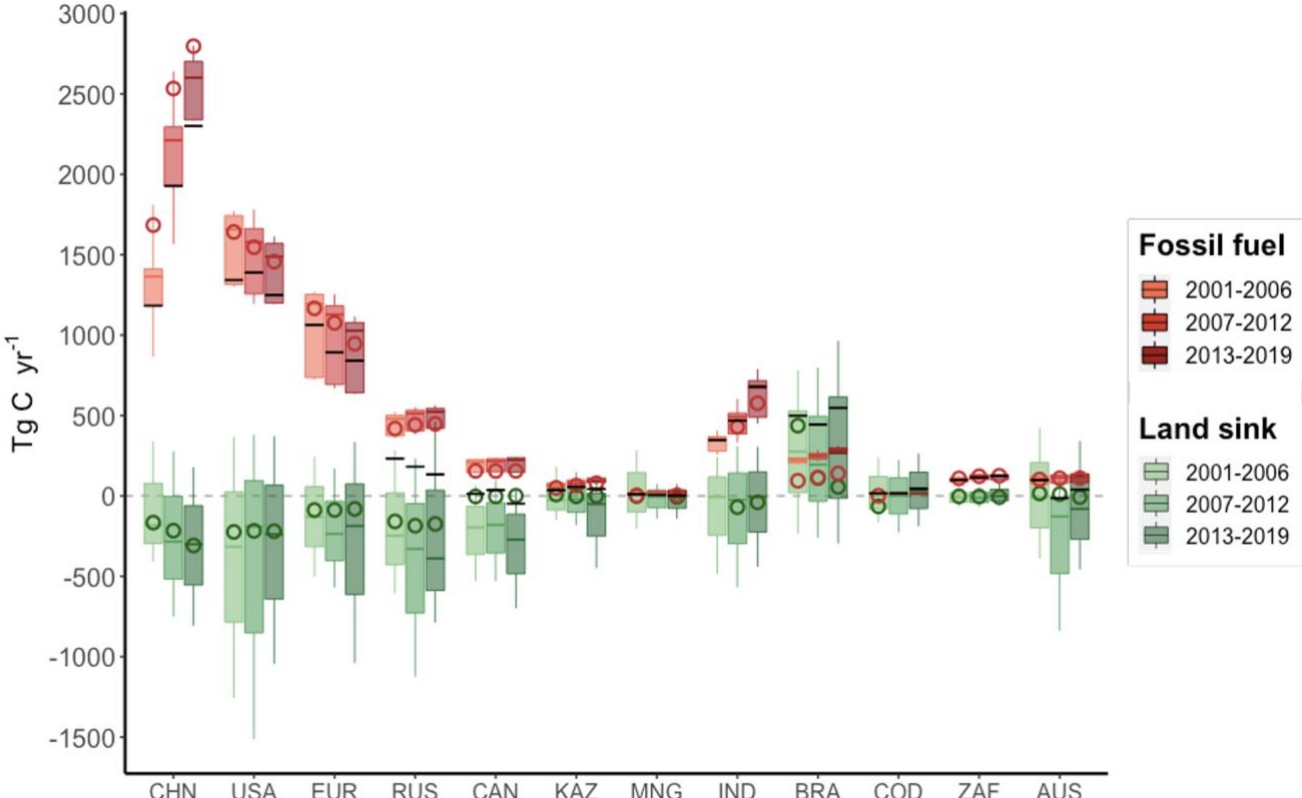

**Figure 8. CO₂ fluxes for land sink, and fossil fuel and cement emissions. Box plots represent inversions for different time periods. Horizontal lines inside box plots denote median values for inversions. Vertical lines of box plots denote minimum and maximum values for selected time periods. Black horizontal lines symbolize net CO₂ fluxes (fossil fuel and cement emissions + land flux) for inversions. Circumferences represent the mean of UNFCCC inventory data available for each time period.**


Figure 8 presents fossil fuel and cement emissions, the net land CO₂ flux and the net flux from the sum of the land flux and fossil emissions. Fossil CO₂ emissions are obtained by re-aggregating to national totals the emission maps provided by each modeling group. These emissions are not optimized by inversions and may differ from UNFCCC inventories because: most non-Annex I countries do not have annual emissions estimates and inversions use as fixed priors gap-filled annual data from

CDIAC and BP statistics for non-Annex I countries from (Friedlingstein et al., 2020), whereas the red circumferences in Fig 8 show the average of emissions from available BUR or NC reports in each period (thus, for each period only data of available years of national inventories are used to calculate the average). For Annex I countries, the prior fossil CO₂ emission maps prescribed to the inversions from Jones et al. (2021) match by construction to national totals from UNFCCC. For non-Annex I countries, the prior fossil CO₂ emission map is based mainly on CDIAC national emissions (Friedlingstein et al., 2020).

Therefore, the data presented in Fig 8 show differences between NI data and inversion priors, in particular for EUR, possibly



due to: 1) international bunker fuel emissions from ships and aviation counted in inversions as surface emissions but not included in UNFCCC national registers, 2) interpolation or regridding by inversion modelers of gridded fossil fuel emissions, or 3) our aggregation of national emissions from coarse resolution inversions back to national totals (see section 1.3). Residual differences have been corrected when presenting in Fig 8 the land sink to match the emissions of Friedlingstein et al. (2020)

but these inconsistencies of emissions between inventories and what is prescribed as prior fossil emissions in inversions should be kept in mind for future studies. There is a striking consistency in Fig 8 between means for each 6-7 yr period of UNFCCC land carbon stock changes and inversions of net land $CO_2$ fluxes, after the adjustments presented in section 1, for all major emitters. Although the range of land carbon inversions is large compared to the small uncertainties estimated by NI reports, the median value of inversions is within 37% of NIs for China (CHN), USA, India (IND) and Brazil (BRA). Inversions give a

larger sink compared to UNFCCC data for several countries, such as Russia (RUS, 85%), EU27 & UK (EUR, 113%), Australia (AUS, > 600%), Canada (CAN, >10,000%). The underestimation by NIs could stem from underestimated soil carbon storage change (many of them assume that e.g. grassland soils are neutral whereas this is probably not the case), or from prescribing bunker fuel aircraft emissions at the surface in the transport models of inversions, which will imply a larger compensatory land $CO_2$ sink. For CHN, given the large amount of fossil emission, there is an aliasing effect between the choice of a prior emission

estimate and the magnitude of the inferred national land sink (Saeki and Patra, 2017). The most interesting differences are for CAN and AUS where non-managed lands, and the lack of accounting of disturbance emissions and recovery $CO_2$ sink by inventories may explain systematically lower sink estimates in inventories compared to inversions. CAN uses relatively old growth functions that do not capture much of the recent transient impact of environmental changes such as rising $CO_2$ and longer growing seasons. AUS considers all forest as formally "managed" but the vast majority (100 Mha of 'other native forest')

are assumed in carbon equilibrium in its national inventories, thus with this assumption there is no biomass loss because of the wood removals is zero.

In this study, we applied a mask of unmanaged forests to inversions gridded $CO_2$ fluxes, which has mainly the effect to reduce $CO_2$ uptake over the countries that have a large fraction of unmanaged forests, namely by 14% in RUS, 30% in CAN, 16% in

BRA and 16% on COD. Brazil is a specific case because although large fractions of the Amazon forest are slightly disturbed by 'management' activities, it contains a significant fraction of protected areas and Indigeneous territories (23% of the total forest area in BRA) (Alejo et al., 2021; IWGIA, 2021) which are counted as managed land, by a political decision of land use. Thus, there is a large mismatch between nationally reported areas of unmanaged land (316 Mha in 2010 according to Table 3.109 from MCTI (2016)) and the intact forest mask we used (166 Mha in 2010, ~33% of the national forest cover). According

to Supp. Table 3 of Grassi et al. (2021), the share of intact forest over total forest was around 40% in CAN and BRA, and 20% in RUS. This share depends on the threshold used to define forest, but in BRA, our intact forest area (16%) used to exclude the inversion fluxes from unmanaged land may be too small. In comparison, the land removals from NI are compiled based on the national communication reports of Brazil that says for instance that about half of their forest is unmanaged. We also note that we did nothing to mask unmanaged grasslands and rangelands for $CO_2$ fluxes, even though these systems are thought to





be larger $CO_2$ sinks per unit area than managed grasslands (Chang et al., 2021a). In the future, it should be possible to mask inversions using the area simulated as managed grasslands by Chang et al. (2021a), that is, 1650 Mha or a fraction of 33% of the global grassland area from their study. The masking of inversions' fluxes over unmanaged forests in USA, CHN and EUR has negligible impact on the net land $CO_2$ flux from inversions given their small /negligible share of forest being declared as unmanaged.


Regarding the effect of $CO_2$ fluxes caused by lateral processes, which do not result in national carbon stock changes, the correction that we applied to inversions (section 1) is equivalent to reproducing the rules of accounting by countries, where wood products harvested (also biofuels) are considered to be emitted where they have been produced, even though these products can be exported and $CO_2$ emitted elsewhere. On the other hand, domestically produced and consumed wood, which

is the majority of wood use in most countries, will induce subnational patterns of $CO_2$ sources and sinks, assumed to be captured by inversions, and not considered explicitly here as an inversion adjustment. To our knowledge, no country is accounting for carbon in traded crop products (as it is not a stock change) nor for carbon transferred from soils to rivers and outgassed, buried in aquatic sediments or transfered to the ocean. In inventories, observed soil carbon stock changes should implicitly include carbon leached or eroded from soils. However, since very few inventories are based on actual soil carbon

change estimates, but rather use assumptions or models that ignore the river loop of the carbon cycle, it is possible that the amount of carbon remaining in soils is overestimated by these approaches (Lauerwald et al., 2020). We found that altogether, the correction of inversions by $CO_2$ fluxes induced by the lateral transport from river, crop and wood products goes from a net source of 19 TgC yr$^{-1}$ in Sudan (mainly crop import) to a net sink of 169 TgC yr$^{-1}$ in Brazil (mainly lateral export). The river correction always makes the inversion net land $CO_2$ flux a smaller sink, whereas the trade of crop and wood can be a net $CO_2$

source or a sink, depending on the balance of exports and imports. We found that these trade fluxes are a source of $CO_2$ in net importing countries China, EU27 & UK, Japan and a $CO_2$ sink in wood and food exporting countries like the USA, Brazil, Argentina. We outline the fact that most of the carbon lost by soils to rivers in a country is outgassed in territorial waters (see section 1.3). Inversion results should partly include this source, although without prescribing it in an explicit manner (e.g. in their prior), but part of this $CO_2$ source could also be mis-allocated to other countries in the flux increment of inversions. The

same remark holds true for $CO_2$ fluxes induced by crop and wood products growth, harvest and trade. Although there is uncertainty about the share of unmanaged land as well as the lateral fluxes, we still make our efforts to narrow the gap between NI and inversions by excluding flues in unmanaged land and adjusting lateral fluxes from inversions. However, more profound and systematic analysis and comparisons is called to harmonize the different scopes between national inventory compilations and inverse models.






## 5.2 Land CO$_2$ fluxes as a function of the agricultural land use ratio

Following Janssens et al. (2005), we regressed national estimates of land CO$_2$ fluxes per area (from inversions) against an agricultural land-use ratio, AR, defined as the ratio of cropland and grasslands area to the sum of cropland, grassland and forest land of each country. The areas of different land use types are extracted from the FROM-GLC global land cover dataset (Gong et al., 2019). The expectation is that croplands and grasslands are small sources or sinks of CO$_2$ compared to forests, so that a country with a greater AR should be a smaller sink or a larger source. Janssens et al. (2005) found such a relationship across EU countries. In the data presented in Fig 9, we see indeed that all temperate and boreal countries (filled circles in Fig 9, excluding EU countries) where all the land is managed show a strong positive relationship fitting by a parabola function (R$^2$=0.6) between the land CO$_2$ fluxes per area and AR. Thus, AR appears to be a good first order predictor of the CO$_2$ balance of these countries, altogether representing 73% of the global land sink of CO$_2$ by inversions. RUS and CAN have a low AR ratio of ≈ 0.2 but the land CO$_2$ sink per area is significantly smaller in RUS. Both RUS and CAN have comparable sink densities (in managed lands) than temperate countries, according to inversions, despite smaller boreal forest growth rates and shorter growing seasons. The reason why RUS managed forests have a smaller sink per area than CAN could reflect inaccuracies in our mask of managed land compared to national data, and possibly, higher disturbances in RUS over managed forests. For tropical forested countries, there is a decoupling between AR and the land CO$_2$ flux per area being a net CO$_2$ source, with deforestation rates being the predominant driver of CO$_2$ fluxes (in particular COD and BRA). In the EUR, the data from inversions show France (FRA) and Germany (DEU) as outliers with larger sinks, actually larger than the NIs (~300% for FRA and ~750% for DEU in the 2010s), probably due to the coarse resolution of inversions, with several models adjusting CO$_2$ fluxes over larger regions than those two individual countries, thus scaling their priors instead of constraining each grid box within EUR countries.

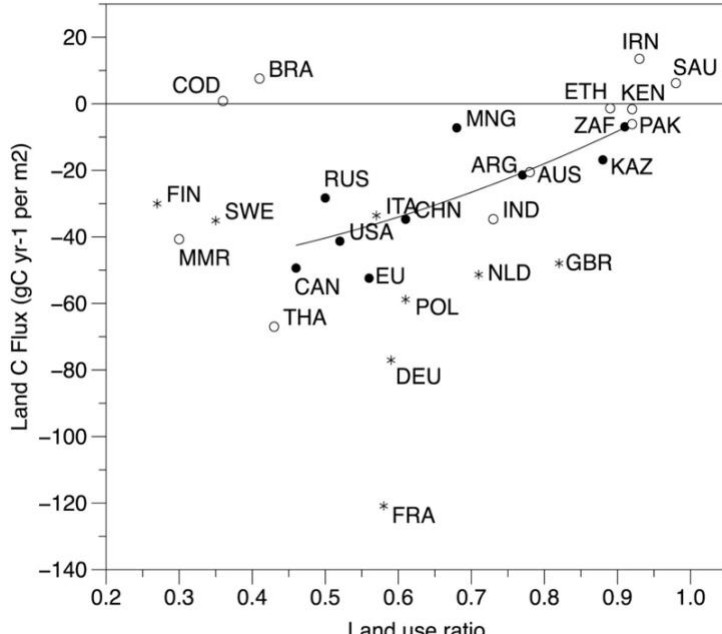

**Figure 9. Annual land CO₂ flux from inversions per unit of total land area versus the agricultural land-use ratio (AR),**
**defined as the ratio of cropland and grasslands area to the sum of cropland, grassland and forest land of each country.**
**Areas of different land use types in all countries are calculated based on FROM-GLC** (Gong et al., 2019)**. Open circles**
**are for subtropical, tropical, and southern hemisphere countries and filled circles for temperate and boreal countries.**
**The black horizontal line indicates zero land flux per area; the black curve denotes the relationship between land**
**carbon flux per area and AR by fitting a parabola function for temperate and boreal countries (filled circles). For**
**information, additional EU countries are shown by star marks (FIN Finland, SWE Sweden, DEU Germany, FRA**
**France, NLD The Netherlands, UK United Kingdom, ITA Italy, POL Poland). The land C flux is the median of the six**
**inversions in the 2010s, after the adjustment given in Equation (1). The high land use ratio for SAU reflects the FROM-**
**GLC land cover dataset classification of mostly desertic areas as grasslands.**



## 5.3 Uncertainties due to the separation of natural from anthropogenic CH₄ emissions

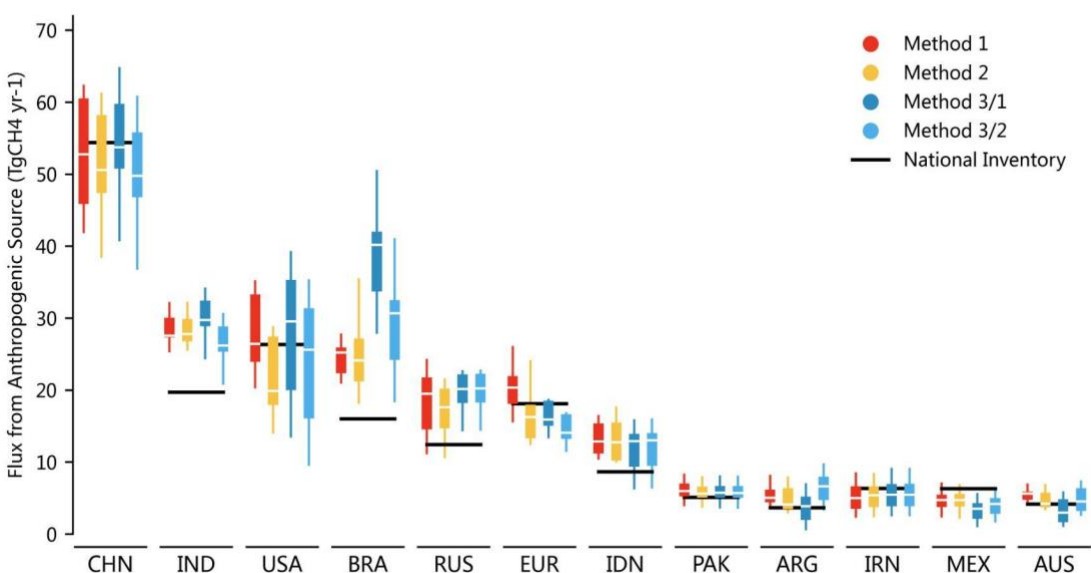

**Figure 10. Interquartile and min, max of total anthropogenic CH₄ emissions separated using different methods based on GOSAT inversions in the 2010s and the total anthropogenic CH₄ emissions from national inventories. For each region, vertical boxes show median, interquartile range, and min/ max of the GOSAT inversions. Each color represents a different separation approach, as defined in section 1.4. Black lines denote the average of total anthropogenic CH4 emissions from national inventories in the 2010s with available reported years.**


Uncertainty of anthropogenic CH₄ emissions using the inversion method is suggested by the spread between models (due to transport models and other inversion specific settings) but also from the method chosen to separate anthropogenic from natural emissions. The data shown in Fig 10 compare the results of the four different separation methods presented in Section 1. It shows that the uncertainty due to the separation method is generally much smaller than the spread between different models,

derived from the fact that inversion vertical ranges appear large relative to differences due to separation procedure. In China (CHN), the between-method range of median inversion estimates of anthropogenic emissions is 4 TgCH₄ yr⁻¹, compared to the mean model spread (interquartile) of 46~60 TgCH₄ yr⁻¹. The range between the median of different methods is 10 TgCH₄ for USA, 6 TgCH₄ for EU27 & UK (EUR) and 3 TgCH₄ for Russia (RUS). In Brazil (BRA), methods 3/1 and 3/2 based on ecosystem models of wetland emissions to diagnose natural emissions give a much larger anthropogenic emissions. This is

likely because these models underestimate natural emissions (Sawakuchi et al., 2014), e.g. do not have emissions from flooded forests (Pangala et al., 2017), from the open river itself, from palm swamps and peat complexes (Winton et al., 2017), which





are an important source in BRA (Melack et al., 2004). The method 1 which is based on the original separation of natural vs anthropogenic emissions from inversions is not systematically different from other methods, but their results differ markedly due to their different use of data sources in each natural/anthropogenic part (see section 1.4). Method 1 gives a slightly higher

anthropogenic emission than other methods for EUR by 4 TgCH$_4$ yr$^{-1}$ averagely, by 3 TgCH$_4$ yr$^{-1}$ for USA and a smaller value than other methods for RUS by 1 TgCH$_4$ yr$^{-1}$. The positive difference between method 1 chosen for this study and other methods (Fig S5) imply a better match of inversions with respect to NI for the EU if other methods were used. For the USA, if we were using method 2, the anthropogenic CH$_4$ emissions would be smaller by 2 TgCH$_4$ yr$^{-1}$, which would further accentuate the underestimation of inversion emissions compared to the NI, especially before 2010. For RUS, even if other

methods were used to compare with NI, our statement about an underestimation of anthropogenic CH$_4$ emission by the NI report remains valid. Importantly, in CHN, our result that inversions produce systematically smaller anthropogenic CH$_4$ emissions than NI data is also robust to the choice of method.

**5.4 Contributions from fossil fuel versus agriculture and waste sectors in CH$_4$ emissions**

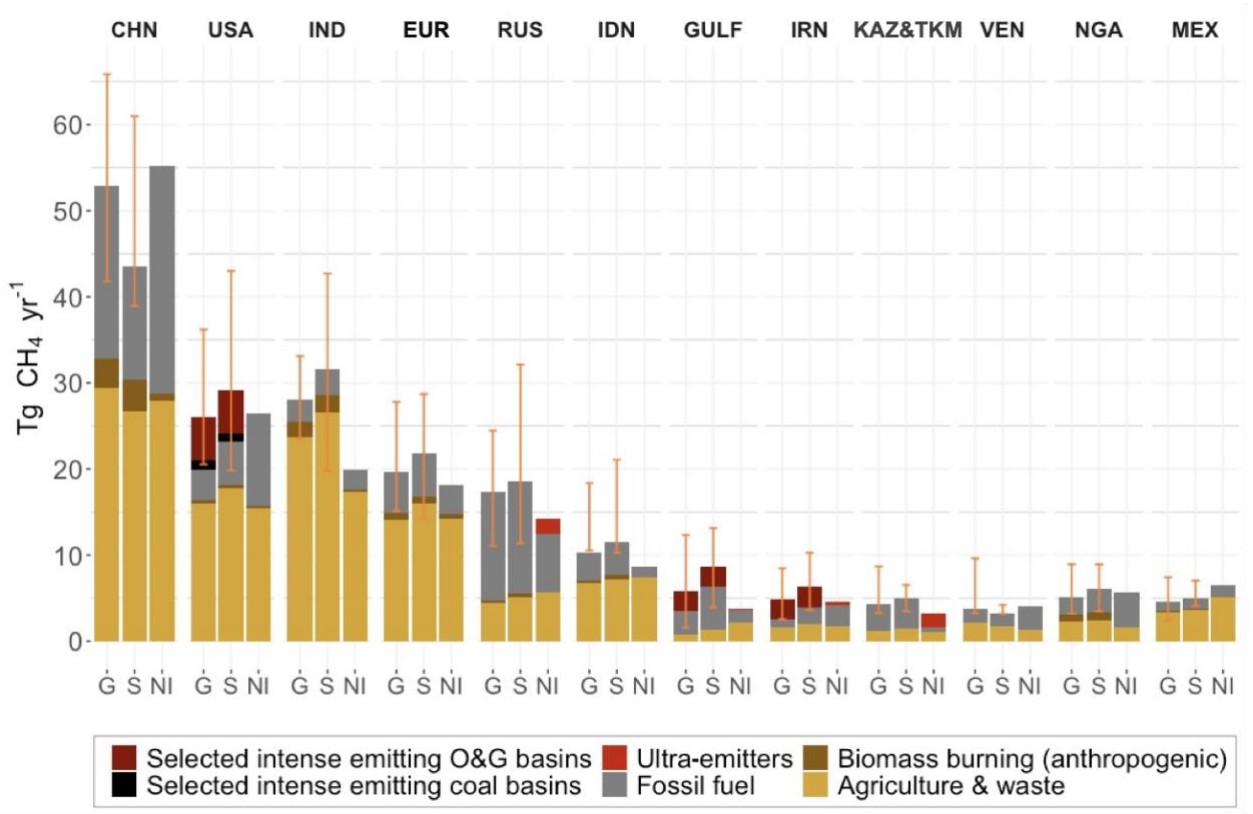


**Figure 11. Anthropogenic CH$_4$ emissions from in-situ (S) and GOSAT (G) inversions compared to national inventories (NIs). S and G data correspond to the mean of inversion medians from the last 5 years of the available inversion data**



**(2013 to 2017). Error bars denote minimum and maximum values for S and G inversions. NI values represent the mean**
**of the three most recent available country reports during the period 1999-2017. Dark red and black emissions represent**
**the fraction of fossil fuel emissions from intensely emitting oil and gas (O&G) basins and from intensely emitting coal**
**basins, respectively, derived from the KAYRROS Global Methane Watch (Fig S7 and S8; Table S3). On top of NI**
**emissions, emissions from ultra-emitters (red) are added to NI estimates (diagnosed from S5P-TROPOMI**
**measurements for the period 2019-2020, (Lauvaux et al., 2021)). For the countries where individual basin emissions are**
**shown, the grey bar is the rest of fossil emissions, i.e. the inversion fossil fuel emissions minus the sum of basins'**
**emissions. Anthropogenic biomass burning was estimated by subtracting GFED emissions (van der Werf et al., 2017)**
**excluding emissions over agricultural lands from the total "biomass burning" emissions of each inversion (Method 1 in**
**section 1).**

Fig 11 compares the share of the different sectors for anthropogenic $CH_4$ emissions across a selection of 12 top emitting
countries (selected countries from top anthropogenic and top fossil $CH_4$ emitters). Generally, inversions partition agricultural
and waste emissions consistently with NIs within the respective uncertainties of both approaches. Inversions provide, however,
larger biomass burning emissions than reported by NIs, partly because we assumed that all biomass burning and biofuel
emissions were anthropogenic in inversion results whereas countries report only fire emissions on managed lands, and
emissions of biofuel burning used for house heating and cooking. Inversions tend to produce higher $CH_4$ emissions than NI
for all oil and gas major emitting countries (except CHN), in particular the USA, Russia (RUS) and Kazakhstan &
Turkmenistan (KAZ&TKM). This under-reporting (also discussed in section 3.2) can be due to the fact that inventories and
emission factors do not consider $CH_4$ leaks from ultra-emitting events consisting of very large and sporadic emissions, like
accidental leaks (Cusworth et al., 2018). Here, we used the first global quantitative estimate of ultra-emitters derived by
Lauvaux et al. (2021) from S5P-TROPOMI measurements, namely all short duration leaks from oil and gas facilities (e.g.
wells, compressors) with an individual emission > 20 $tCH_4\,h^{-1}$, each event lasting generally less than one day. Using the event
duration obtained by fitting a local dispersion model to observed S5P-TROPOMI methane plumes, all ultra-emitting events in
each country were aggregated during the period from January 2019 to December 2020 (Table S3). Assuming that those large
leaks are not reported by NIs, they were added to national reports of fossil $CH_4$ emission as red stack bars in Fig 11. Doing so
reduces the misfit with inversions, especially in RUS and KAZ&TKM. Ultra-emitters represent 319% (1.6 $TgCH_4\,yr^{-1}$) of NI
fossil emissions for KAZ&TKM, 25% (1.7 $TgCH_4\,yr^{-1}$) for RUS and 4% (0.06 $TgCH_4\,yr^{-1}$) for the countries grouped in the
GULF region. We also considered emissions derived from S5P-TROPOMI measurements at the scale of regional extraction
basins for oil, gas and coal. Four major oil and gas basins were considered (Fig S7) as specific areas where many individual
wells and storage facilities are concentrated, each of them with a probability of emitting $CH_4$, and forming a clear regional
enhancement of $CH_4$ detected in S5P-TROPOMI imaged and assimilated with a regional inversion into a regional $CH_4$
emission budget (see Table S3, Supplementary text and KAYRROS Methane Watch, https://www.kayrros.com/methane-
watch/). Such basin scale emissions were diagnosed from regional inversions using S5P-TROPOMI atmospheric



measurements. Here, we assumed that those basins are already counted as part of the national $CH_4$ budgets from in-situ and GOSAT inversions. Thus, they are shown here for the share of total national fossil emissions that they represent (dark red bars part of total fossil emissions in inversions results displayed in Fig 9. In the USA, the Permian basin emissions represent between

54% (in-situ, S) and 61% (GOSAT, G) of the total national fossil $CH_4$ estimates from inversions. Alone, the Permian basin contributed 16% of the total gas and 35% of the oil extracted in the US in 2019 (EIA, 2021a). Our average 2019-2020 emission estimate in the Permian basin is 2.3 $TgCH_4$ $yr^{-1}$ from S5P-TROPOMIdata, which is consistent with an estimate of 2.7 $TgCH_4$ $yr^{-1}$ from O&G industries in the Permian Basin reported by Zhang et al. (2020) but contrasts with the 1.4 $TgCH_4$ $yr^{-1}$ emission estimate for the entire USA reported by EPA (2020). In the GULF, emissions from the basin comprising Iraq and Kuwait

represent 29.8% (S) - 44.8% (G) of the total estimated fossil emissions of this region. This basin estimation encompasses four of the highest oil-producing fields in the world and its oil production accounts for 31.5% of all the countries in the GULF region (EIA, 2020). The basin estimation from inversions for IRN (2.3 $TgCH_4$) represents 55.2% (S) - 71.0% (G) of fossil fuel estimated emissions and  94.8% of the national total NI report.

**5.5 Overlooked importance of natural $N_2O$ emissions in non-Annex I countries**

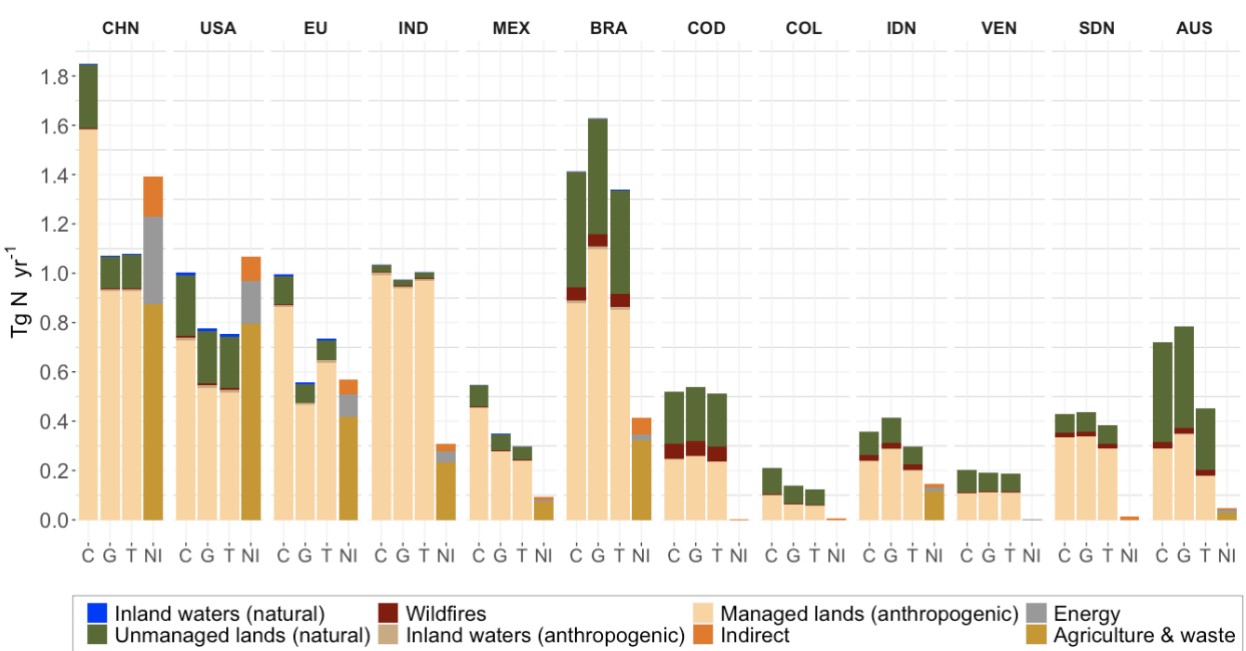

**Figure 12. $N_2O$ emission fluxes in TgN $yr^{-1}$ for CAMS (C), GEOS-Chem (G) and INVICAT (T) inversions compared to NI data for 2010-2016. C, G, and T data correspond to the mean of fluxes from 2010 to 2016. NI values represent the mean of available data for the same period. Anthropogenic rivers emissions are from Yao et al. and they are represented**

**for information as part of managed land emissions, as they are captured by inversions (section 1). Natural river**



**emissions are considered as the average of 1900-1910 (section 1) which are removed from total emissions of inversions. Data of wildfires correspond to GFED reported values for the period of interest** (van der Werf et al., 2017)**. They are counted as non-anthropogenic emissions following equation (6) in section 1, and reported here for information.**

As shown in section 4, the estimation of $N_2O$ emission fluxes by emission inventories is challenging and currently some non-Annex I countries (e.g. COD) have no estimates available. Figure 12 compares inversions (CAMS, GEOS-Chem and INVICAT) to available NI reported $N_2O$ emission fluxes. Because emission sectors of NIs and inversions are limited and do not coincide with each other, the comparison of N2O emission sectors between these two data sources can only be accomplished partially. The main innovation proposed in this study has been to separate total inversion fluxes into unmanaged

and managed lands so that the emissions over managed lands minus the natural inland water emissions can be compared with inventories (section 1). We can see in the data presented in Fig 12 that natural emissions (natural fluxes from lakes and rivers plus fluxes from unmanaged lands) account for 32% of the mean inversions total for BRA, 47% in COD and 57% in AUS. In temperate industrialized countries with a smaller fraction of unmanaged land, the magnitude of natural fluxes relative to anthropogenic ones is smaller. In comparison to emissions from unmanaged lands, the natural emissions from rivers are always

of a very small magnitude. In general, removing natural emissions tends to improve the agreement with inventories in non-Annex I countries. In Annex I countries, it tends to make inversion-based emissions smaller than NIs. The main uncertainty is on the area of grasslands and forests assumed to be unmanaged from our masks, and how well they correspond to the unmanaged areas used by each country. A large area of extensively grazed land e.g. in Mongolia and Kazakhstan is considered here as natural whereas those countries may consider them as being under management, even though the nitrogen cycle and

$N_2O$ emissions are close to natural conditions for extensive grazing. The consistent pattern of higher emissions in inversions than NI among the three model inversions for non-Annex I countries, suggests that emission factors for these countries may be revised upwards, and that indirect emissions may be re-assessed and reported when it is not the case (Table S2). On the other hand, for USA and CHN, the median of inversion emissions is smaller than inventories and CAMS is higher than the two other inversions considered. Concerning smaller inversion estimates for CHN and USA compared to NI, this could be

because the Tier 1 used by NI assumes static EFs whereas EFs may change (smaller or larger) depending on cropland Nitrogen Use Efficiency (NUE) and climate. USA has improved its NUE in the 1990s compared to the 1980s considerably (Lassaletta et al., 2014) but if the EFs used are based on flux measurements in the 1980s these could be too high. A recent data driven model of direct cropland $N_2O$ emissions (Wang et al., 2020b) using non-linear EF and regional N-fertilizers input data, found emissions smaller than Tier 1 methods, which would be in better agreement with inversions. Another source of uncertainty in

the $N_2O$ inversions is the prior estimates for land versus ocean. Since the ocean is not well constrained in the inversions, having a too high ocean prior will mean the land total will be underestimated and vice versa.



### 5.6 Comparison with regional emissions

Table 3 compares the results of the global inversions used in this study, with regional inversion results compiled from the literature, generally obtained with higher resolution regional transport models and sometimes using atmospheric data not assimilated by global inversions. Global inversion results are given without and with adjustments for $CO_2$ fluxes due to lateral transport, and for anthropogenic emissions estimated using equation Method 1 for $CH_4$ and equation (6) for $N_2O$ (section 1). The purpose of introducing a $CO_2$ flux correction (section 1.3) was to make an accurate comparison with inventories, but since regional $CO_2$ inversions did not use such an adjustment, here we focus on comparing regional inversions with global ones 1075 without adjustment.

For $CO_2$ fluxes in CHN, except for the large uptake found by the inversion Wang et al. (2020a), all previous regional inversion results fall within the range of our global inversion ensemble for their period of overlap, indicating no systematic bias of global inversions. Note that Wang et al. (2020a) is a global inversion using new Chinese stations data and a discretization of fluxes 1080 into smaller sub-regions within China. In BRA, the range of global inversions also covers regional inversion results, yet with global inversions being a small $CO_2$ source in 2010 (194 TgC yr$^{-1}$) and regional inversions a large source in that year. The regional inversions from Gatti et al. (2014) and Van der Laan-Luijk et al. (2015) used aircraft $CO_2$ and CO profiles in the Amazon also give a larger flux change between 2010 (a dry year, a $CO_2$ source in the Amazon) and 2011 (a wet year, a $CO_2$ sink in the amazon) than global inversions that do not assimilate these aircraft data. For the EUR, the range of regional 1085 inversions (Petrescu et al., 2021b) is similar to the one of global inversions. For North America, however, regional inversions give a higher average $CO_2$ uptake than global ones, yet within their range.

For $CH_4$ emissions in CHN, the results of all regional inversions (Miller et al., 2019; Thompson et al., 2015) are consistent with our global inversion ensemble, although Miller et al. (2019) is in the upper range. For BRA or the Amazon basin, 1090 interestingly, regional inversions (Tunnicliffe et al., 2020; Wilson et al., 2016) provide systematically higher $CH_4$ emissions than global in-situ inversions estimates, but regional inversions include wetlands and rivers, which can explain their higher values. If removing natural emissions from regional inversions, then their values would be consistent with global results, i.e. Tunnicliffe et al. (2020) estimated that $CH_4$ emissions from BRA is 33.6±3.6 TgCH$_4$ yr$^{-1}$, with 19.0±2.6 TgCH$_4$ yr$^{-1}$ from anthropogenic sources, falls within the range of of our estimates for anthropogenic emissions from global inversions (19 - 36 1095 TgCH$_4$ yr$^{-1}$). In the EU and the US or North America, the regional inversions in Table 3 which have higher resolution transport models give higher $CH_4$ emissions than global inversions, even when only anthropogenic emissions are considered in regional inversions. This suggests that global models may systematically underestimate $CH_4$ emissions from those two high emitters. For $N_2O$, we have several regional inversions for North America, all producing higher emissions than the median of global inversions by a factor of four, on average. The only regional $N_2O$ inversion over Europe is also about two times higher than 1100 the median of global inversions.





**Table 3. Comparison of global inversions in this study with regional inversions from the literature (the range from the inversion ensemble is given in parentheses, unless stated otherwise). Values in bold text show as statistically valid that the regional inversion results fall within the range of our global inversion ensemble. \* = estimates for the Amazon Basin. \*\* = 10th and 90th percentiles. \*\*\* = The separation of anthropogenic emissions from regional emissions excludes wetlands but uses different methods than in this study. \*\*\*\* = no adjustment of regional CO₂ inversion results was performed, unlike in column 5 and based on Eq (1) for global inversions. † = GOSAT ACOS and OCO2 ACOS XCO2 products** (Kong et al., 2019; GES DICS, 2021)**. ^ = for China, only two NI reports are available in 2005 and 2010 and the average of the two years is given in the table.**

$CO_2$ (TgC yr$^{-1}$)

| Region | Results from published literatures | | | This study | | |
|---|---|---|---|---|---|---|
| | Period | Regional inversions \*\*\*\* | References | Median of global inversions (all lands without adjustment) | Median of global inversions (managed lands with adjustment) | National inventories from UNFCCC |
| China | 2010-2016 | −1110±380 (in-situ) <br> -1070 ± 330 (in-situ+GOSAT-CO2 †) <br> -880 ± 430 (OCO2-ACOS) | (Wang et al., 2020a) | -273 <br> (-712 to -23) | -245 <br> (-635 to 1) | -247 |
| | 2001-2010 | **−330** <br> **(-290 to -640)** | (Zhang et al., 2014) | **-279** <br> **(-509 to -6)** | -216 <br> (-431 to 50) | -222 |
| | 2006-2009 | **−450±250** <br> **(-390 to -510)** | (Jiang et al., 2016) | **-394** <br> **(-666 to -216)** | -334 <br> (-586 to -165) | -222 ^ |
| | 2002-2008 | **-280±180** | (Jiang et al., 2013) | **-305** <br> **(-476 to -37)** | -238 <br> (-392 to 23) | -166 |
| Brazil | 2010 | 480±180 \* | (Gatti et al., 2014) | 9 <br> (-403 to 416) | 194 <br> (-241 to 539) | 71 |



| Region | Year | Results from published literatures | Reference | | | |
|---|---|---|---|---|---|---|
| | 2011 | 60±100 * | (Gatti et al., 2014) | -58 (-347 to -9) | 101 (-224 to 223) | 48 |
| | 2010 | 460±320 * | (Alden et al., 2016) | 9 (-403 to 416) | 194 (-241 to 539) | 71 |
| | 2011 | 180±320 * | (Alden et al., 2016) | -58 (-347 to -9) | 101 (-224 to 223) | 48 |
| | 2012 | -210±320 * | (Alden et al., 2016) | 234 (-457 to 331) | 328 (-195 to 501) | 13 |
| | 2010 | 70±420 to 310±420 * | (van der Laan-Luijkx et al., 2015) | 9 (-403 to 416) | 194 (-241 to 539) | 71 |
| | 2011 | -150±420 to -270±420 * | (van der Laan-Luijkx et al., 2015) | -58 (-347 to -9) | 101 (-224 to 223) | 48 |
| EU27+ UK | 2006-2015 | **−381 to -138** | (Petrescu et al., 2021b) | **-231 (-465 to 119)** | -202 (-444 to 127) | -88 |
| | 2004 | **-570** | (Schuh et al., 2010) | **-909 (-2036 to -140)** | -300 (-834 to 165) | -101 |
| North America | 2000-2005 | **-650 (-400 to -1000)** | (Peters et al., 2007) | **-833 (-1614 to -79)** | -251 (-648 to 137) | -114 |
| | 2003 | **-970±210** | (Deng et al., 2007) | **-851 (-1474 to -37)** | -334 (-546 to 58) | -119 |
| | 2004 | **0 to -1000** | (Gourdji et al., 2012) | **-909 (-2036 to -140)** | -300 (-834 to 165) | -101 |

CH$_4$ (TgCH$_4$ yr$^{-1}$)

| Region | Results from published literatures | This study |
|---|---|---|





| | Period | Regional inversions total net emissions | Regional inversions, anthropogenic emissions *** | References | Median of global inversions anthropogenic (Method 1) | National inventories from UNFCCC |
|---|---|---|---|---|---|---|
| **China** | 2000-2011 | 44±3.5 | | (Thompson et al., 2015) | in-situ: 40 (34 to 49) | 50 |
| | 2010-2015 | 59 | **57** | (Miller et al., 2019) | **in-situ: 45 (37 to 61)** **GOSAT: 52 (40 to 62)** | 54 |
| | 2015 | 61.5±2.7 | | (Miller et al., 2019) | in-situ: 44 (37 to 62) GOSAT: 53 (45 to 65) | NaN |
| **Brazil** | 2010-2018 | 33.6 ± 3.6 | **19.0±2.6** | (Tunnicliffe et al., 2020) | **in-situ: 24.0 (19.1 to 35.7)** **GOSAT: 24.7 (20.8 to 32.0)** | 16.0 |
| | 2010 | 36.5-41.1 * | | (Wilson et al., 2016) | in-situ: 26.2 (22.7 to 33.3) GOSAT: 25.9 (20.7 to 38.7) | 15.5 |
| | 2011 | 31.6-38.8 * | | (Wilson et al., 2016) | in-situ: 23.0 (17.6 to 33.1) GOSAT: 23.1 (20.8 to 29.0) | 15.8 |
| **EU27+UK** | 2006-2012 | 31.1 (29.3 to 32.7) | **25.8 (24.0 to 27.4)** | (Petrescu et al., 2021a) | **in-situ: 21 (15 to 30)** | 20 |
| | 2006-2012 | 26.8 ( 20.2–29.7)** | | (Bergamaschi et al., 2018) | in-situ: 21 (15 to 30) | 20 |
| **USA** | 2007-2008 | | 44.5±1.9 | (Miller et al., 2013) | in-situ: 29 (22 to 44) | 28 |
| | 2010-2015 | 42.4 (37.0 to 42.9) | **30.6 (29.4 to 31.3)** | (Maasakkers et al., 2021) | **in-situ: 29 (18 to 45)** **GOSAT: 26 (20 to 36)** | 26 |





| Region | Period | | | | | |
|---|---|---|---|---|---|---|
|  | 2009-2011 | 51.3-52.5 | **40.2-42.7** | (Turner et al., 2015) | **in-situ: 29 (17 to 46)** | 27 |
|  | 2004 | 37.0±1.4 | **30.1±1.3** | (Wecht et al., 2014) | **in-situ: 26 (19 to 36)** | 27.5 |
| **North America** | 2009-2011 | 88.5-91.3 | | (Turner et al., 2015) | In-situ: 35 (18 to 53) | 31.7 |
|  | 2003 | 49 | | (Kort et al., 2008) | in-situ: 30 (25 to 42) | 32.4 |

**N$_2$O (Tg N$_2$O yr$^{-1}$)**

| Region | Results from published literatures | | | | This study | |
|---|---|---|---|---|---|---|
|  | Period | Regional inversions, total | Regional inversions, anthropogenic | References | Median of global inversions (Anthropogenic) | National inventories from UNFCCC |
| **EU27+UK** | 2005-2014 | 1.5 | 1.5 | (Petrescu et al., 2021a) | 0.84 (0.75 to 0.97) | 0.58 |
| **North America** | 2003 | 4.3 | | (Kort et al., 2008) | 0.94 (0.71 to 1.16) | 1.08 |
|  | 2004 | 2.5 | | (Kort et al., 2010) | 0.75 (0.41 to 1.10) | 1.10 |
|  | 2008-2014 | 2.5±0.5 | | (Nevison et al., 2018) | 0.96 (0.81 to 1.12) | 1.05 |
|  | 2004-2008 | 3.3-4.1 | | (Miller et al., 2012) | 0.91 (0.70 to 1.12) | 1.05 |

**Data availability**

GHG (CO$_2$, CH$_4$, N$_2$O) data from inverse models and UNFCCC national inventories are available at https://doi.org/10.5281/zenodo.5089799 (Deng et al. 2021).



This dataset contains 5 data files, including GHG data from inverse models and UNFCCC national inventories in the top emitter countries:

- **CO₂_inversion_1990-2019**: annual $CO_2$ flux from 6 inversion models in three sectors:

- 'land flux (all land)' -> land flux from all land
- 'land flux (managed land)' -> land flux from managed land
- 'land flux (managed land + lateral adjustment)' -> land flux from managed land by adjusting the lateral flux

- **CH₄_inversion_2000-2017**: CH4 flux from 10 in-situ inversion (2000-2017) and 11 satellite inversion (2010-2017) models from four sectors:

- 'anthropogenic (method x)' -> anthropogenic emissions from managed land. x could be 1, 2, 3.1 and 3.2, representing different methods to calculate the emissions in this sector:
- 'fossil' -> emissions from the fossil sector
- 'agriculture & waste' -> emissions from the agriculture and waste sector combined
- 'biomass burning' -> emissions from biomass burning

- **N₂O_inversion_1997-2016**: anthropogenic $N_2O$ emissions from 3 models.

- **Inventory_1990-2019**: inventory data collected from UNFCCC national inventories. The classification of sectors is corresponding with the inversion data files for each gas species.

- **Inventory_1990-2019_IPCC**: inventory data collected from UNFCCC national inventories in the IPCC category.

**Conclusions**

One step forward of this study is that it proposes a new toolkit of methodologies to improve the consistency between inversions and UNFCCC inventories for each of the three greenhouse gases, by post-processing inversion results to make them comparable with the rules of accounting of inventories. For $CO_2$, we separated the fluxes in unmanaged land by using the intact forest mask and separated the fluxes coupled to lateral transport (by river or by trade) from inversions; For $CH_4$, we proposed three methods to split the anthropogenic fluxes from inversions by aggregating prior estimates from each sector or by removing fluxes of natural processes; For $N_2O$, we also separated the fluxes from managed land by using the same method on $CO_2$ and account for the indirect $N_2O$ emissions. In the case of $CO_2$, using a mask of managed lands is also critical for large forested countries (Grassi et al., 2021), and tents to make their "carbon sink" smaller than when using inversion fluxes over all the grid cells. Here we made a first attempt to use an intact 'non managed' forest mask for this purpose. Such a mask could be extended to unmanaged grasslands in future studies, e.g. following recent work by Chang et al. (2021a). Here, we recommend that countries should report their managed land in a spatially explicit manner to enable a better evaluation of national emission reports using inversions (and other observation based approaches), and countries should also follow the recommendations of the IPCC 2006 Guidelines encouraging countries to use atmospheric data as an independent check on their national reports (Eggleston et al., 2006; Buendia et al., 2019) (see also the discussion in Chevallier (2021)). Removing





from inversions the CO2 fluxes coupled to lateral transport, which represent no carbon stock change and are not all visible to

inventories, generally make the "carbon sink" significantly smaller in northern mid latitude countries (e.g. -26% lower in CAN and 18% lower in RUS), than if raw inversion data were used. All harvest is seen by NI as a loss of carbon stock in forest. Then, the wood that remains in the country enters the Harvested Wood Products (HWP) pool (where gains and losses are recorded). What is 'invisible' in NIs is the wood that enters the HWP pool of a foreign country. $CH_4$ and $N_2O$ emissions have been even less explored for a systematic comparison of inversions with inventories. For these two gas species, we improved

the processing of inversion gridded fluxes to separate anthropogenic fluxes from the total emissions, in order to provide estimates that can be compared with NIs for policy implementation. For $CH_4$, we proposed three methods to remove the signal of natural $CH_4$ emissions, and found that their robustness is country dependent, the separation of natural emissions to retrieve anthropogenic emissions being more difficult in countries that have both large natural and anthropogenic emissions and few atmospheric stations, like RUS or BRA. We certainly recommend here to reduce the uncertainty of prior estimates and improve

estimations of natural sources using e.g. better bottom-up datasets of wetland area, rivers and lakes, and their $CH_4$ emissions rates, in order to make further in-depth comparisons between these methods. For $CH_4$, a second notable result is that despite the large spread of inversions, both in-situ and GOSAT inversions show valid differences with NI anthropogenic emissions. We also found that Kazakhstan and Turkmenistan in central Asia and the Gulf countries in the Middle East, characterized by oil- and gas-producing industries, report much less $CH_4$ emissions than atmospheric inversions. It is fair to say that in this

region, there are few ground stations, and inversions could depend on their prior fluxes, but the fact that GOSAT and in-situ data point to NI emissions being underestimated suggests areas for future research to constrain the emissions of these countries. We recommend here to develop regional campaigns (such as those performed in Alvarez et al. (2018)), to refine emission factors, and to track regional oil, gas and coal basins emissions and ultra-emitter site level emissions using new tools (such as moderate and high-resolution satellite imagery). For $N_2O$, the prevalence of large tropical natural sources, being outside the

responsibility of countries if they are located on unmanaged lands, has been overlooked before. For example, nearly half of the forests in Brazil are unmanaged according to its national inventory report. We did not solve this problem, but highlighted it and proposed a new method to remove natural emissions from inversion total emissions. As many non-Annex 1 countries which will have to produce inventories for the global stocktake are tropical countries with a very active nitrogen cycle and large natural $N_2O$ emissions, a decoupling will exist between targeted emissions reductions and the observed growth rate of

$N_2O$: it may hamper the eventual effectiveness of mitigation policies, that are directly reflected in the UNFCCC national reports, especially for this greenhouse gas.

The study of global inversions at the country scale rather than at the traditional subcontinent scale (e.g. the "Transcom3 regions" of Gurney et al. (2002)) obviously pushes inversions close to the limit of their domain of validity, even in the case of

large countries. The densification of observation networks and systems, especially from space, increases the observational information available at all spatial scales and gradually makes it possible to study smaller countries. This densification must be accompanied by a corresponding increase in the horizontal resolution of inversion systems (both the transport model and



the control vector to be optimized). Note that the spatial resolution of most inverse models such as those contributing to the global carbon/methane/nitrous oxide budget is larger than 1 degree (see Table A4 in Friedlingstein et al. (2020), Table S6 in Saunois et al. (2020), and Supp. Table 18 in Tian et al. (2020)). They will likely soon have to go below one degree on a global scale to remain competitive for this type of study, despite the high computational challenge posed by the atmospheric inversion of long-lived tracers.

## Acknowledgements

This synthesis has been co-funded by the European Space Agency Climate Change Initiative ESA-CCI RECCAP2 project (ESRIN/ 4000123002/18/I-NB) and by the European Commission Prototype system for a Copernicus $CO_2$ service (CoCO$_2$, grant agreement no. 958927) and Observation-based system for monitoring and verification of greenhouse gases (VERIFY, grant agreement no. 725546). The authors are very grateful to the many data providers (measurements, models, inventories, atmospheric inversions, hybrid products, etc.) that are directly or indirectly used in this synthesis.

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
