# Peer review of "Comparing national greenhouse gas budgets reported in UNFCCC inventories against atmospheric inversions"

_Earth System Science Data, 2021_

## Referee Comment (RC2)

[referee-annotated manuscript omitted]

---

## Author Comment (AC1)

RC1: 'Comment on essd-2021-235', Anonymous Referee #1, 20 Sep 2021

This is important and long-time awaited paper, describing the methodology and results of making inversion modeling comparable with GHG inventories in the UNFCCC national reporting. The paper provides multidimensional assessment, which considers three major gases: CO2 (managed and unmanaged land), CH4 (anthropogenic emissions, fossil, agriculture & waste) and N2O (anthropogenic), separately for large counties.

The paper provides motivation to different communities and countries to advance the modeling and reporting: Inverse modeling community – to check the reasons for inconsistency between the models and with other estimations; independent validation of country UNFCCC reporting; upscaling in situ measurements; etc.

The advantage of inversions is that they provide insights on seasonal and interannual greenhouse gas fluxes anomalies, e.g. during extreme events such as drought or wildfire, while national inventories tend to average and delay with recording emissions.

The paper is well written, all data processing steps are described, the results are discussed extensively. I have just a few comments.

Line 143: "we chose countries with an area that contains at least 13 grid boxes of the highest resolution grid-scale inversions"? Any reason for such a decision? Was it a pre-condition or did you find out minimum number of pixels (13) after country selection process?

We thank the Reviewer for the comment. The inversion models usually have coarse spatial resolution. The CAMS CO2 and N2O inversions, for instance, solve for fluxes on the grid of their transport model, with cells of 1.875° in latitude by 3.75° in longitude; the Jena-Carboscope CO2 inversion has a resolution of 4° by 5° which is about two times coarser. Some other inversions solve for fluxes at a coarser scale than their transport model, over small or large regions. Given this discretization of fluxes in global inversions, small countries (e.g. below the size of a middle-sized EU country like France) cannot be resolved well. Therefore, we focused the comparison on large countries or grouped countries that are large emitters (for each gas) and cover an ad hoc area of $\approx$ 13 CAMS pixels, roughly 1.25 M km$^2$. An exception is Venezuela for CH4 that is just below this limit (916,400 km$^2$).

Line 229: "intact forest areas (that are unmanaged, by definition)". Definitions of managed forest are different in different thematic areas and vary in different countries for UNFCCC reporting.

IPCC Guidelines (2006) defines "Managed land is land where human interventions and practices have been applied to perform production, ecological or social functions". For example, intact forest in a national park is managed to support ecological functions (i.e. the forest is under fire protection). This intact forest is considered as "managed" for UNFCCC reporting. Based IBFRA analysis (unpublished IBFRA report, 2021), 49% of forest area in the IFL - Intact Forest Landscapes (Potapov et al., 2017) polygons belongs to "managed land" according to UNFCCC national reporting in Boreal biome. At the same time substantial amount of "unmanaged forest" are outside of IFL polygons, e.g. northern open

woodlands. I understand that in absence of global dataset of managed land the IFL is a logical compromise. However, the readers should be warned about this limitation.

We thank the Reviewer for the comment and the information about the unpublished IBFRA report. We have added and addressed this limitation in Line 1149-1154: "However, it should be noted that there are discrepancies between the Intact Forest Landscapes maps of Potapov et al. (2017) that we use and the unmanaged land defined in NGHGIs. For instance, an intact forest protected in a national park will be classified as managed in the corresponding NIR. Conversely, some areas of "unmanaged forest" are outside of the Intact Forests defined by Potapov et al. (2017), e.g. northern open woodlands"

Before that, we have also discussed the limitations of the approach we used to extract flux from unmanaged land in Line 232-235:
"This approach assumes that non-intact forest represents a reasonably good proxy of managed forest reported in the NIRs (Grassi et al., 2021). In the absence of a machine-readable definition of the areas considered to be managed in many NIR, this choice remains somewhat arbitrary and other unmanaged land datasets could have been used (Ogle et al., 2018; Chevallier 2021)."

And we also discussed the mismatch of CO2 fluxes from NIRs and inversions in Brazil and Canada in Line 877-887:

"Brazil is a specific case because although large fractions of the Amazon Forest are slightly disturbed by management activities, it contains a significant fraction of protected areas and Indigenous territories (23% of the total forest area in BRA) (Alejo et al., 2021; IWGIA, 2021) which are counted as managed land, by a political decision on land use. Thus, there is a mismatch between nationally reported areas of unmanaged land (316 Mha in 2010 according to Table 3.109 from MCTI (2016)) and the intact forest mask we used (166 Mha in 2010, ~33% of the national forest cover). According to Supp. Table 3 of Grassi et al. (2021), the share of intact forest over total forest was around 40% in CAN and BRA, and 20% in RUS. This share depends on the threshold used to define forest, but in BRA, our intact forest area (16%) used to exclude the inversion fluxes from unmanaged land may be too small, which means that BRA has non-managed land in non forest biomes. In comparison, the national communication reports of Brazil report that about half of the forest area is unmanaged."

---

## Author Comment (AC2)

RC2: 'Comment on essd-2021-235', Anonymous Referee #2, 05 Oct 2021

I find this manuscript to be a fundamental milestone in addressing a key need in developing independent methods for monitoring country reporting of GHG emissions to the UNFCCC. The possibility to use inversion model results as an independent, science-based tool for monitoring has been long put forward by the IPCC, so much so that teh refined 2019 guidelines dedicated new sections to it. A few countries in Europe have even begun including early systems in their national GHG inventory (NGHGI) processes.

Having said that, in fact becasue of it, my opinion is that this manuscript, while offering a view into what is possible currenlty with available data and model capabiliies, needs to be equally fully transparent about the underlying uncertainties and limitations. I have made many comments throughuot the manuscript that point to these needs, with recommendations to the authors to address each of them.

More in general, the authors need to be transparent about the follwoing issues:

1. While comparisons with NGHGI data appear to be remarkably positive in the sense of demonstrating the power of inversion modeling, I was left with the doubt that at least some of this agreement is built in and a consequence of significant calibration. For one, the inversions are driven by primers, typically global model data which in turn are often derived from the NGHGIs data --or are constricted in a similar fashion. The reader should be informed of the degree of depence between one (the primer, consistent with and oftern adjusted to reproduce NGHGI) and the other (the results, compared to NGHGI).

We agree with the reviewer that it is important to beware of dependence between the datasets. However, we would like to point out that NGHGIs are independent from inversions for land $CO_2$ fluxes, as we explain below. Atmospheric $CO_2$ inversions use prior values either from null fields (Jena Carboscope) or from gridded land biosphere models which do not represent management and that are not tuned to reproduce NIRs. Inversions also use sufficiently large prior errors to match atmospheric observations. Therefore, the comparison is not circular. For $CH_4$, the prior of inversions is based on global emission datasets (see Saunois et al. 2020) produced by the scientific community. For anthropogenic emissions, the inversions rely mostly on different versions of EDGAR inventory (here mostly v4.2 and v4.3.2), which is not produced based on NGHGI, though they may share the same sources of activity data for some countries, EDGAR apply its own global TIER approach accounting for specific emission factors, technology dependency and spatialization process. Further, $CH_4$ inversions use prior fluxes for wetlands, other natural sources, and fire emissions based on Earth Observation and land surface models. These fluxes are not reported by NGHGIs, which makes the prior settings of $CH_4$ inversions further independent from NGHGIs, when using them to retrieve anthropogenic emissions from total emissions. For $N_2O$, inversions are used as prior global inventories for anthropogenic emissions, again based on different emission factors and activity data than NGHGIs, and additional datasets (e.g. ocean and land models) for other sources. This has been clarified in the revised manuscript.

2. Even when the independence of model and observation are sufficiently demonstrated, the reader is still poorly informed on the degree of uncertainty upon which the inversion modeling depends upon and the implications for result interpretation. Uncertainty in input primer data; uncertainty in land cover maps used to derive land fluxes; uncertainty in lateral flows used to modify apparent inversoin signals; among several others.

We have added a paragraph about uncertainties from inversions. Deriving uncertainties for individual inversions can be challenging depending on the approach used (especially the 4Dvar), and they are often not provided. Having said that, considering an ensemble of models and inversions allow to account for the uncertainty on transport error, prior emissions (at least for $CH_4$ where inversions have used different priors). Thus, by presenting "uncertainty" as the min-max range of the inversions, we hope we are able to capture most of the "uncertainty" and at least account for uncertainty due to the choice of a different transport model for each GHG. Uncertainty on lateral fluxes was not formally estimated and arises from FAO data which contain (small) systematic errors, plus a small uncertainty on the carbon content of commodities.

3. The reader is not sufficiently informed of the mapping that was used to "read" the UNFCCC country data used for assessing the goodness of the inversions. As the authors state, Annex I data are pretty straightforward. But the same is not true for GHG data from NAI countries. How were in practice LULUCF, forest land and other type land data read and mapped into categories that are instead used by the inversion models? Such information should at least appear in the SI, but it's not there.

We thank the Reviewer for this comment. In Table 1, we illustrated how we processed the land flux from inversions. To supplement our processing of NGHGIs data from non-Annex I countries, we added a table in SI to show how the sectors under different classification systems (IPCC 1996/2006) are mapped (grouped) into the aggregated sectors we defined in this study. Table S3. IPCC category systems defined by the two IPCC guidelines (IPCC, 1997, 2006)

| Non Annex I | | Annex I |
|---|---|---|
| IPCC 1996 | IPCC 2006 | CRF (IPCC 2006) |

| 1. Energy | 1.A Fuel Combustion - Sectoral Approach | 1.A.1 Energy Industries
1.A.2 Manufacturing Industries and Construction
1.A.3 Transport
1.A.4 Other Sectors
1.A.5 Other (Not elsewhere specified) | 1 ENERGY | 1A Fuel Combustion Activities | 1A1 Energy Industries
1A2 Manufacturing Industries and Construction
1A3 Transport
1A4 Other Sectors
1A5 Non-Specified | 1. Energy | A. Fuel combustion (Reference approach / Sectoral approach) | 1. Energy industries
2. Manufacturing industries and construction
3. Transport
4. Other sectors
5. Other |
|---|---|---|---|---|---|---|---|---|
| | 1.B Fugitive Emissions from Fuels | 1.B.1 Solid Fuels
1.B.2 Oil and Natural Gas | | 1B Fugitive Emissions from Fuels | 1B1 Solid Fuels
1B2 Oil and Natural Gas
1B3 Other Emissions from Energy Production | | B. Fugitive emissions from fuels | 1. Solid fuels
2. Oil and natural gas and other emissions from energy production |
| | | | | 1C Carbon Dioxide Transport and Storage | 1C1 Transport of CO2
1C2 Injection and Storage | | C. CO2 Transport and storage | 1. Transport of CO2
2. Injection and storage
3. Other |
| 2. Industrial Processes | 2.A Mineral Products | 2.A.1 Cement Production
2.A.2 Lime Production
2.A.3 Limestone and Dolomite Use
2.A.4 Soda Ash
2.A.5 Asphalt Roofing
2.A.6 Road Paving with Asphalt
2.A.7 Other | 2 INDUSTRIAL PROCESSES AND PRODUCT USE | 2A Mineral Industry | 2A1 Cement Production
2A2 Lime Production
2A3 Glass Production
2A4 Other Process Uses of Carbonates
2A5 Other (please specify) | 2. Industrial processes and product use | A. Mineral industry | 1. Cement production
2. Lime production
3. Glass production
4. Other process uses of carbonates |
| | 2.B Chemical Industry | 2.B.1 Ammonia Production
2.B.2 Nitric Acid Production
2.B.3 Adipic Acid Production
2.B.4 Carbide Production
2.B.5 Other | | 2B Chemical Industry | 2B1 Ammonia Production
2B2 Nitric Acid Production
2B3 Adipic Acid Production
2B4 Caprolactam, Glyoxal and Glyoxylic Acid Production
2B5 Carbide Production
2B6 Titanium Dioxide Production
2B7 Soda Ash Production
2B8 Petrochemical and Carbon Black Production
2B9 Fluorochemical Production
2B10 Other (please specify) | | B. Chemical industry | 1. Ammonia production
2. Nitric acid production
3. Adipic acid production
4. Caprolactam, glyoxal and glyoxylic acid production
5. Carbide production
6. Titanium dioxide production
7. Soda ash production
8. Petrochemical and carbon black production
9. Fluorochemical production
10. Other |
| | 2.C Metal Production | 2.C.1 Iron and Steel Production
2.C.2 Ferroalloys Production
2.C.3 Aluminium Production
2.C.4 SF6 Used in Aluminium and Magnesium Foundries | | 2C Metal Industry | 2C1 Iron and Steel Production
2C2 Ferroalloys Production
2C3 Aluminium Production
2C4 Magnesium Production
2C5 Lead Production
2C6 Zinc Production
2C7 Other (please | | C. Metal industry | 1. Iron and steel production
2. Ferroalloys production
3. Aluminium production
4. Magnesium production
5. Lead production
6. Zinc production
7. Other |

| | | | | | | | |
|---|---|---|---|---|---|---|---|
| | | | | | specify) | | |
| | **2.D Other Production** | 2.D.1 Pulp and Paper
2.D.2 Food and Drink | | **2D Non-Energy Products from Fuels and Solvent Use** | 2D1 Lubricant Use
2D2 Paraffin Wax Use
2D3 Solvent Use
2D4 Other (please specify) | **D. Non-energy products from fuels and solvent use** | 1. Lubricant use
2. Paraffin wax use
3. Other |
| | **2.E Production of Halocarbons and SF₆** | 2.E.1 By-product emissions | | **2E Electronics Industry** | 2E1 Integrated Circuit or Semiconductor
2E2 TFT Flat Panel Display
2E3 Photovoltaics
2E4 Heat Transfer Fluid
2E5 Other (please specify) | **E. Electronic industry** | 1. Integrated circuit or semiconductor
2. TFT flat panel display
3. Photovoltaics
4. Heat transfer fluid
5. Other |
| | **2.F Consumption of Halocarbons and SF₆** | | | **2F Product Uses as Substitutes for Ozone Depleting Substances** | 2F1 Refrigeration and Air Conditioning
2F2 Foam Blowing Agents
2F3 Fire Protection
2F4 Aerosols
2F5 Solvents
2F6 Other Applications | **F. Product uses as substitutes for ODS** | 1. Refrigeration and air conditioning
2. Foam blowing agents
3. Fire protection
4. Aerosols
5. Solvents
6. Other applications |
| | **2.G Other** | | | **2G Other Product Manufacture and Use** | 2G1 Electrical Equipment
2G2 SF6 and PFCs from Other Product Uses
2G3 N2O from Product Uses
2G4 Other (please specify) | **G. Other product manufacture and use** | 1. Electrical equipment
2. SF6 and PFCs from other product use
3. N2O from product uses
4. Other |
| **3. Solvent and Other Product Use** | | | | **2H Other (please specify)** | 2H1 Pulp and Paper Industry
2H2 Food and Beverages Industry
2H3 Other (please specify) | **H. Other** | |
| **4. Agriculture** | **4.A Enteric Fermentation** | | **3 AGRICULTURE, FORESTRY AND OTHER LAND USE** | **3A Livestock** | 3A1 Enteric Fermentation
3A2 Manure Management | **A. Enteric fermentation** | 1. Cattle
2. Sheep
3. Swine
4. Other livestock |
| | **4.B Manure Management** | | | **3B Land** | 3B1 Forest Land
3B2 Cropland
3B3 Grassland
3B4 Wetlands
3B5 Settlements
3B6 Other Land | **B. Manure management** | 1. Cattle
2. Sheep
3. Swine
4. Other livestock |

| | | | | | | |
|---|---|---|---|---|---|---|
| | **4.C Rice Cultivation** | 4.C.1 Irrigated
4.C.2 Rainfed
4.C.3 Deep Water | **3C Aggregate Sources and Non-CO2 Emissions Sources on Land** | 3C1 Biomass Burning
3C2 Liming
3C3 Urea Application
3C4 Direct N2O Emissions from Managed Soils
3C5 Indirect N2O Emissions from Managed Soils
3C6 Indirect N2O Emissions from Manure Management
3C7 Rice Cultivations
3C8 Other (please specify) | **C. Rice cultivation** | |
| | **4.D Agricultural Soils** | 4.D.1 Direct Soil Emissions
4.D.2 Pasture, Range and Paddock Manure
4.D.3 Indirect Emissions | **3D Other** | 3D1 Harvested Wood Products
3D2 Other (please specify) | **D. Agricultural soils** | |
| | **4.E Prescribed Burning of Savannas** | | | | **E. Prescribed burning of savannas** | |
| | **4.F Field Burning of Agricultural Residues** | 4.F.1 Cereals
4.F.2 Pulses
4.F.3 Tubers and Roots
4.F.4 Sugar Cane | | | **F. Field burning of agricultural residues** | |
| | **4.G Other** | | | | **G. Liming** | |
| | | | | | **H. Urea application** | |
| | | | | | **I. Other carbon-contining fertilizers** | |
| | | | | | **J. Other** | |
| **5. Land-Use Change and Forestry** | **5.A Changes in Forest and Other Woody Biomass Stocks** | 5.A.1 Tropical Forests
5.A.2 Temperate Forests
5.A.3 Boreal Forests | | | **A. Forest land** | 1. Forest land remaining forest land
2. Land converted to forest land |
| | **5.B Forest and Grassland Conversion** | 5.B.1 Tropical Forests
5.B.2 Tropical Savanna / Grasslands
5.B.3 Temperate Forests
5.B.4 Grasslands
5.B.5 Boreal Forests
5.B.6 Grasslands / Tundra
5.B.7 Other | | | **B. Cropland** | 1. Cropland remaining cropland
2. Land converted to cropland |

**4. Land use, land-use change and forestry**

| | | | | | | | | |
|---|---|---|---|---|---|---|---|---|
| | **5.C Abandonment of Managed Lands** | 5.C.1 Tropical Forests
5.C.2 Tropical Savanna / Grasslands
5.C.3 Temperate Forests
5.C.4 Grasslands
5.C.5 Boreal Forests
5.C.6 Grasslands / Tundra
5.C.7 Other | | | | **C. Grassland** | 1. Grassland remaining grassland
2. Land converted to grassland | |
| | **5.D CO₂ Emissions and Removals from Soil** | 5.D.1 Cultivation of Mineral Soils
5.D.2 Cultivation of Organic Soils
5.D.3 Liming of Agricultural Soils | | | | **D. Wetlands** | 1. Wetlands remaining wetlands
2. Land converted to wetlands | |
| | **5.E Other** | | | | | **E. Settlements** | 1. Settlements remaining settlements
2. Land converted to settlements | |
| | | | | | | **F. Other land** | 1. Other land remaining other land
2. Land converted to other land | |
| | | | | | | **G. Harvested wood products** | | |
| | | | | | | **H. Other** | N2O Emissions from Aquaculture Use
CH4 from artificial water bodies | |
| **6. Waste** | **6.A Solid Waste Disposal on Land** | 6.A.1 Managed Waste Disposal on Land
6.A.2 Unmanaged Waste Disposal Sites | **4 WASTE** | **4A Solid Waste Disposal** | 4A1 Managed Waste Disposal Sites
4A2 Unmanaged Waste Disposal Sites
4A3 Uncategorised Waste Disposal Sites | **5. Waste** | **A. Solid waste disposal** | 1. Managed waste disposal sites
2. Unmanaged waste disposal sites
3. Uncategorized waste disposal sites |
| | **6.B Wastewater Handling** | 6.B.1 Industrial Wastewater
6.B.2 Domestic and Commercial Wastewater | | **4B Biological Treatment of Solid Waste** | | | **B. Biological treatment of solid waste** | 1. Composting
2. Anaerobic digestion at biogas facilities |
| | **6.C Waste Incineration** | | | **4C Incineration and Open Burning of Waste** | 4C1 Waste Incineration
4C2 Open Burning of Waste | | **C. Incineration and open burning of waste** | 1. Waste incineration
2. Open burning of waste |
| | **6.D Other** | | | **4D Wastewater Treatment and Discharge** | 4D1 Domestic Wastewater Treatment and Discharge
4D2 Industrial Wastewater Treatment and Discharge | | **D. Wastewater treatment and discharge** | 1. Domestic wastewater
2. Industrial wastewater
3. Other (as specified in table 5.D) |

| 7. Other | | | 5 OTHER | 4E Other (please specify) | | 6. Other (please specify) | E. Other | |
|---|---|---|---|---|---|---|---|---|
| | | | | 5A Indirect N2O Emissions from the Atmospheric Deposition of Nitrogen in NOx and NH3 | | | | |
| | | | | 5B Other (please specify) | | | | |
| Memo Items | International Bunkers | Aviation Marine | Memo Items | International Bunkers | International Aviation International Water-borne Transport | Memo Items | International Bunkers | Aviation Marine |
| | CO₂ Emissions from Biomass | | | Multilateral Operations | | | Multilateral operations | |
| | | | | CO2 from Biomass Combustion for Energy Production | | | CO2 emissions from biomass | |
| | | | | | | | CO2 captured | For domestic storage For storage in other countries |
| | | | | | | | Long-term storage of C in waste disposal sites | |
| | | | | | | | Indirect N2O | |
| | | | | | | | Indirect CO2 | |

4. Which were the global datasets used as primers? THis is also not clear. For LULUCF, it's unclear whether FAOSTAT was used, as a complement to country data or not.

As explained above, priors of inversions for LULUCF CO2 are independent from NGHGIs. For LULUCF CO2, in the countries shown in the paper, we used data from CRF reports, NC and BUR. We did not use the data from Grassi et al. who completed NGHGIs by FAO in some instances. The Grassi et al. estimates were used only for comparing the total managed land CO2 flux from inversion with "national reports". We have clarified that this global estimate of LULUCF CO2 sink is not strictly from UNFCCC data as it contains FAO data in some countries.

Other data used to make inversions comparable with NGHGIs are described in Section 1.3. For CO2, we subtract the CO2 fluxes induced by "lateral transport" processes across the border of each country from the inversion net land CO2 flux. These data come from Ciais et al. (2020b) for river exports and wood trade, and were calculated based on FAOSTAT and Xu et al., 2021 for crop trade. For CH4, datasets from termites, wetlands, freshwater (lakes and reservoirs), geological processes and wildfires are included to separate the anthropogenic flux from the total net flux. Prior estimates for anthropogenic methane emissions are from EDGAR (different versions were used depending on the inversions). For N2O, we removed the pre-industrial freshwater flux (from a model, assuming that the anthropogenic freshwater flux is counted under NGHGIs indirect emissions) and wildfires fluxes from the net flux over managed land.

5. Considering the above, I found that the authors tended to discuss discrepancies between inversion model results and NGHGI by consistently assuming that the models were right and the NGHGI wrong. Although some of the theoretical reasoning may at times be correct, the underlying uncertainty in both sources would in my opinion advice for greater caution in drawing such assumptions--- in general it does not seem that any definitive direction can be deduced from the available data.

We thank the Reviewer for the comment. We fully agree that we shouldn't make any assumption on whether inversion data are more accurate or not than NGHGIs, or whether NGHGIs data are wrong or not. And we do not make this assumption because both NGHGIs and inversions have pros and cons. Our goal is to assess and try to understand the discrepancies between inversions and NGHGIs, as the truth remains unknown. Such discrepancies can provide insights about sources of uncertainties in national budgets and allow identifying areas where either NGHGIs or inversions can be improved. From the perspective of NGHGIs, we show that countries, especially the non-Annex I countries should improve the completeness and transparency of all sector data under the IPCC guidelines, adopt country-specific emission factors in the calculations, and that all countries with a large area of managed land should report this area as a spatially explicit dataset to enable a more robust comparison with inversions. From the side of inversions, finer spatial resolution, better priors, and more stations in tropical regions are needed for more accurate outputs. To avoid possibly biased language, we revised a few sentences in the manuscript and replaced "under-reporting" which conveys the idea of a deliberate under-reporting by "lower estimation", e.g:

"In India (IND), both the in situ and GOSAT inversions also give a higher anthropogenic emission than the NGHGIs data and the share of emissions from natural wetlands is much smaller than in RUS and BRA, reducing the risk of aliasing anthropogenic for natural emissions, and suggesting a lower estimation by the NGHGI"

"Possible reasons for lower anthropogenic emissions for nearly all the non-Annex 1 countries can be the use of Tier 1 emission factors (EF) which may be lower than when soil and climate dependence is accounted for (Philibert et al., 2013; Shcherbak et al., 2014; Wang et al., 2020b), and the non-linear observed concave response of cropland soils emissions as a function of added N fertilizers (Zhou et al., 2015) which makes emissions higher than the linear relation used by NGHGIs in Tier 1 approaches. "

6. Considering the importance that is being placed--rightfully--on the use of inversion modeling as an additional, independent and very much perfectable instrument in coming years for monitoring the quality of NGHGI data, I would have expected a more detailed and nuanced discussions where current limitations (ie uncertainties but not only--issues of land use definitions are also very important to the usability of such methods) lie, and what a honest assessment of the performance of the current exercise suggests for the future: where are the most important areas for improvement and what can be done to improve the system. For instance, is the currently uncertainty sufficient for use in monitoring GHG mitigation actions? If the inversion models have a given uncertainty range attached to them currently, what is then the minimum range of monitoring that they could permit? IN practice, can a system that carries uncertainties of 50% and above be able to monitor changes in NGHGI inventories (needed to demonstrate mitigation actions) of 10-30% over the next decade?

We broadly agree that inversions as a method still bear too large uncertainties to "verify" or "falsify" national reports. At best, inversions can show if NGHGIs are consistent with independent atmospheric information, which is already valuable since inversions allow, at the same time, constraining global budgets in a consistent manner. Our results show that for CO2, inversions allocate land fluxes with the total being consistent with the observed CO2 growth rate, whereas the sum of NGHGIs gives a too small global land sink. This reconciliation is fundamental for understanding where NGHGIs might be missing carbon sinks, and our results suggest that this missing sink is mainly in boreal countries.

In addition, in some countries such as oil and gas producing countries in the Gulf area and Central Asia, possibly Russia, the results of our CH4 comparison suggest that NGHGIs emissions are significantly lower than estimated by all inversions. The same is true for N2O in tropical countries, Mexico and Australia. Here, inversions despite their uncertainties may detect issues with NGHGIs.

Regarding the large uncertainty of the mean inversion flux, it does not affect trends much, as trends are constrained by accurate measurements of changes in the CO2 growth rate at different stations. Most inversion errors are systematic, but the temporal component of this bias is smaller than the mean bias value. Therefore, at coarse spatial scale, inversions are valuable to monitor trends and the effectiveness of mitigation efforts.

We added sentences in the final paragraph to explain this and added statements about how inversions could be improved.

For all the above reasons, I strongly feel that this manuscript should be published in ESSD, but only after major revisions that address my points.

We hope that our revisions have addressed the valuable comments raised in your comments

Comments in the attachment:
Line 38: this sentence could be re-written for improved clarity. The authors mean simply "national ghg inventories for the land use, land use change and forestry (LULUCF) sector. Co2 only is not correct since they are also looking into CH4 and N2O

We revised the sentence to:
"In support of the Global Stocktake of the Paris Agreement on Climate change, this study presents a comprehensive framework to process the results of an ensemble of atmospheric inversions in order to make their Net Ecosystem Exchange (NEE) carbon dioxide (CO2) flux suitable for evaluating National Greenhouse Gas Inventories (NGHGIs) submitted by countries to the United Nations Framework Convention on Climate Change (UNFCCC)."

Line 44: according to which accounting? and is the uptake only anthropogenic or total (including natural)

We calculate the percentage contributions according to the median of inversion data we used in this studys.

Line 44: use LULUCF acronym, as long as it has been introduced earlier-- as per previous comment

Revised.

Line 48: within NGHGIs -- also acronym to be introduced earlier

Revised.

Line 53: inventories is a generic term, Once introduced as an acronym, better to use NGHGI throughout

Revised.

Line 83: introduce NGHGI acronym here

Revised.

Line 85: I would also add here, once only, that such QAQC processes endorsed by the IPCC also include validation exercises using available global databases, such as from IEA, FAO, etc.

We revised the sentence to: "The IPCC guidelines for NGHGIs encourage countries to use independent information to check on emissions and removals (IPCC, 1997, 2006), such as comparisons with independently compiled inventory databases (e.g. IEA, CDIAC, EDGAR), or with atmospheric concentration measurements interpreted by atmospheric inversion models (see Section 6.10.2 in IPCC (2019))."

Line 95: what is "geographic europe"? I would simply write "europe" at this level of detail

We revised it to "Europe".

Line 121: again: introduced the concept of NGHGI early on -- than once you use the term it's always clear you are talking about anthropogenic fluxes to compare to the inversions.

Thanks. We revised the term throughout the manuscript.

Line 141: could you specify how these percentage contributions were computed? The NGHGI data is incomplete. The inversions data are the object of this analysis. What else was then used?

We calculate the percentage contributions according to the median of inversion data we used in this studys.

Line 149: I think, considering the declared support to the stock take exercise, that the order should be reversed: how do inversion models compare to NGHGI data?

Thanks, we revised it to: "1) how do inversion models compare with NGHGIs for the three gases?;"

Line 167: NIRs are submitted biennally

NIRs are submitted annually by Annex I, non-Annex I submit BURs / BRs

Line 229: not true. The IPCC land use definition allows for a wider definition of "managed", i.e. onne that includes administrative arrangements for things other than wood and pulp production. For instance, national parks in natural state (intact) can be considered managed.
According to current data availability, it's impossible to have the areas of unmanaged mask for each nation. So in this study, we define the intact forest as unmanaged land. We explained it more detail in the next sentences: "This approach assumes that non-intact forest represents a reasonably good proxy of managed forest reported in national GHG inventories (Grassi et al., 2021). In the absence of a machine-readable definition of the plots considered to be managed in many NGHGIs, this choice remains somewhat arbitrary and other unmanaged land datasets could have been used (Ogle et al., 2018; Chevallier, 2021). "

Line 240: please explain in more detail why NEE are not counted in NGHGI. I am not sure that is true, especially when a country (most do) adopt the so-called land use proxy, which in essence allows to label as anthropogenic all fluxes on a given piece of managed land

Only NEE from unmanaged lands are masked in this study.

Line 266: FAOSTAT references should be given as, for example: FAO, 2021. FAOSTAT Database, XYZ domain (plus web link to specific domain, not general FAOSTAT front page), FAO, Rome. Downloaded XYZ.

Done, thanks.

Line 275: what about carbon (and nitrogen) inputs by livestock back into the grasslands they graze?

Not considered in this study.

Line 313: they should not be reported--however many NGHGI likely also are unable to disaggregate natural from anthropogenic fires. On the other hand, certain fires, especially peatland fires, are definetly anthropogenic and are reported in certain BURs, namely Indonesia--which has the most substantial amounts.

We agree, thanks.

Line 319: biomass burning data that mimic NGHI reporting is available in FAOSTAT. Was it used and if not why not

Here we only used the GFEDv4 dataset in this study , to remove the estimate for large scale biomass burning, which is not included in NGHGI, but is in the emissions from the inversions. Also this is the data set used as prior in almost all inversion, so using GFED4 to remove biomass burning emissions allows consistency.

Line 357: That is incorrect. N2O emissions in NGHGIs is directly computed from manure deposition, form livestock, including on natural grasslands and other rangelands

According to IPCC 1996/2006 guidelines, N2O emissions from non-managed lands are not reported in the NGHGIs. Here we assumed the intact forest area and lightly grazed grasslands as non-managed land. Thus, N2O fluxes from these areas are assumed to be not counted in the inventory as well. Although the compilation of manure management in inventories considers the livestock from all land, we made this assumption from the perspective of subtracting the fluxes from unmanaged land from inversion data. We revised this sentence to make it clear: "We assumed the intact forest areas and lightly grazed grassland areas approximate unmanaged land, where the fluxes are not reported in the NGHGIs."

Line 376: as for CH4, FAOSTAT also contains database of N2O emissions from wildfires and managed fires. What was the reason for not using it as an additional source

We think resolving the differences between FAOSTAT and NGHGI is beyond the scope of this paper.

Line 385: This does not seem right. NGHGI contain as per IPCC guidelines indirect N2O emissions from volatilization and leaching. Kindly better explain what is meant here

Some non-Annex countries did not report the AIE emissions in their NGHGIs: "Another important point to ensure a rigorous comparison between inversions and NGHGIs data is whether anthropogenic indirect emissions (AIE) of N2O are reported in NGHGIs reports, even though UNFCCC parties should report these in their NGHGIs according to the IPCC guidelines. For example, South Africa's BUR3 did not report the indirect GHG emissions due to the lack of activity data (DEA, 2019)."

Line 428: this can be a negative number in many developing countries

It could be negative numbers indicating a sink.

Line 442: is this a net sink or a gross one. What was the total sink obtained this way? And if the NGHGI sums are incomplete (the authors state they did not gap gill), then what does a selection based on percentage (partial) global coverage mean?

We calculate this percentage by using the inversion data. We sum the land flux from the 12 countries and compared to the global totals.

Line 449: on these percentages, same question as previous. How were the global total computed?

We calculate the percentage contributions according to the median of inversion data we used in this studys.

Line 465: unclear. The authors previously stated that they made their own compilation of country data to UNFCCC. Now they state they are using Grassi et al.,, which are gap-filled.

We revised the sentence: "While data for the specific countries analyzed in this paper (Fig. 3) are based on our own compilation, the global land use flux from NGHGIs is from Grassi et al. 2021 (based on a compilation of different country submissions to UNFCCC, in few cases gap-filled with country reports to FAO-FRA 2015)."

Line 466: unclear whether this net land flux is only anthropogenic or not.

It's the total net flux on all lands

Line 469: and FAO LULUCF estimates are in fact a net source of some 0.2 Gt C. Were emissions from peat fires included in

this analysis?

We think this really deserves a separate analysis, but not here. NGHGIs do include peat fires (e.g. from Indonesia)

Line 470: possible. However, teh FAOSTAT estimates, which are consistent with existing country data and gap fill the rest within the FAO independent approach, point to a net source, not a larger net sink.

We think resolving the differences between FAOSTAT and NGHGI is beyond the scope of this paper.

Line 472: that is not convincing. Forest inventories upon which the data are largely estimated do track the state of forests over time and should implicitly include all of these impacts.

We delete this sentence.

Line 475: In fact. Also in relation to a previous note, the Harris et al numbers mix anthropogenic and natural fluxes.

We revised this sentence: "Inversions are also smaller than the Tier-1 approach published by Harris et al. (2021), who estimates a sink of 2.1 GtC yr-1 over the last 20 years over managed and unmanaged forests with a large range of ±13 GtC that seems biophysically implausible."

Line 482: this is consistent with similar difficulties in NGHGIs, for which in fact the use of the land use proxy is introduced
Line 484: most countries do report to FAO their carbon stocks over time through the FRA process.
Line 490: it would be interesting to also compare to carbon stock change estimates of FAO, based on FRA data and estimates as emissions/removals in FAOSTAT

Based on our analyses, in most cases NGHGI provide a more reliable/complete source of information on C sink that FAO. FAOSTAT may be useful to gapfill where NGHGI provide no or too old data (many small developing countries). Comparing the data of this paper also with FAOSTAT - while potentially interesting -would extend the original scope and increase the length of the paper. We suggest keeping the original scope. And the Global stocktake will be done with NGHGI data, not with FAO data.

Line 494: the fact that the inversions see a net sink in canada is suspicious. Canada is a large emitter from drained peatlands--according to FAO--so large that the net LULUCF is positive in those estimates.

For CAN, We think the main reason for the difference is that CAN does NOT include most of the recent env. change in the estimated forest sink.

Line 497: is there CO2 from the waste sector? In any case, what does a monitoring of all of these together allow one to do, in terms of QAQC of inventories?

Thanks. The land CO2 flux excludes fossil fuel emissions, and includes the flux mainly from AFOLU.

Line 500: why? I understand this graph is about CO2 only, in which case those from fossil fuel CO2 on the farm is a major flux.

Fossil fuel CO2 on the farm is reported under the category of *Energy* instead of *Agriculture*.

Line 509: so, teh land CO2 flux includes fossil fuels? I would suggest to change the terminology, as in IPCC and in NGHGI inventories by land flux one only understands LULUCF.

Thanks. The land CO2 flux excludes fossil fuel emissions, and includes the flux mainly from AFOLU.

Line 521: this reads like a very soft analysis. Needs some re-writing in my opinion

Thanks. We rewrite this part: "In CHN, the successive national communication estimates in 5 years fall in the range of the six inversions, and give a trend towards an increasing carbon sink. Adjusted inversions provide a median CO2 sink of 142 TgC yr-1 in 2005 and of 245 TgC yr-1 during 2010-2014, consistent with reported values from inventory reports (166 TgC yr-1 in 2005, and an average of 247 TgC yr-1 in 2010, 2012 and 2014). Note that the NGHGIs in 2010 and 2014 reported in China's NC3 and BUR2 used the IPCC 2006 guidelines to calculate the flux from the LULUCF sector, which includes fluxes from six land-use types (forest land, crop land, grassland, wetlands, settlements, and other land). However, the LULUCF sector in the other three years reported in NC1, NC2, and BUR1 only considered fluxes from forest land."

Line 525: what is missing from these detailed discussions is a discussion of uncertainty at the outset. LULUCF fluxes are typically at least 50% uncertain. The inverse models appear to have at least teh same range of uncertainty. So how can one attach any meaning to trends over 5-10 years by declaring that something is increasing or decreasing with any certainty? What remains impressive is the bulk of the estimates, which appear con

According to NGHGI, uncertainty in LULUCF for Annex I countries is usually <50%. For most developing countries, this information is not available. See Supp Info of Grassi et al 2018 for more details https://www.nature.com/articles/s41558-018-

0283-x

Line 533: this is odd, because the NGHGI tend to underestimate emissions from drained peatlands, whereas the inversions should be able to see them-- hence the direction of the sink should be opposite than what seen in the results... the sink should be smaller in the inversions. FAO estimates a large net source for instance

For CAN, I think the main reason for the difference is that CAN does NOT include most of the recent env. change in the estimated forest sink. For RUS, NGHGI may underestimate the sink because of old data.

Line 538: yes but this is a large source of uncertainty between the inversions and the NGHGIs -- countries do not necessarily defined their forests-- and the managed component therein-- according to these academic maps.

Thanks. We agree and discuss it in the conclusion section.

Line 547: please also consider reporting FAO estimates, which are based more closely on country reporting and thus provide a middle way between inversions, GFW and NGHGIs.

Thanks. Comparing FAO estimates might be beyond the scope of our paper. Note that this shows that NGHGI can be quite different/dependent than other global inventories used as prior (FAO, EDGAR, etc).

Line 552: can you also check against the FAO estimates?

Again, comparing with FAO estimates would be an interesting exercise, but in our view a systematic comparison is beyond the scope of the paper.

Line 555: I thought they had greatly increased since 2015?

It remained constant for NGHGIs.

Line 575: that's odd. FAO estimates a large CO2 source over land -- from deforestation.

that's because COD does NOT report data to FAO on change in C stock in forest remaining forests

Line 579: that's why you should also use the independent FAO estimates on this, based on FRA reported data by the country and allowing for a stock change computation

For COD, UNFCCC is indeed poor and contradicting. However, FRA data is not necessarily better. Comparing NGHGI and FRA is a needed exercise, but we think it goes beyond the scope of this paper

Line 582: was this real or how much of this finding is driven by the sue of the GFED as a primer?

The peak was shown by the inversion data in Fig S4.

Line 582: again, this seems to fly in the face of the reality of large emissions from peatland degradation in this area. Where these used form NGHGI or FAO, and how much of their use or not as a primer influences the final results of the inversions

NGHGIs do include peat fires (e.g. from Indonesia)

Line 593: is this pure speculation or is it backed up by carbon cycle model runs to this end?

It's a speculation.

Line 609: are these also the sum of several NGHGI sectors? (ag, waste, etc)

Yes. Anthropogenic emissions from NGHGIs are the sum of energy, industrial process, agriculture, and waste; and biofuel burning

Line 611: what use is there for making such aggregate comparisons in view of the stock take exercise. Aren't these results too much aggregated to be useful? Unless one could use additional models/databases to further disaggregate

This aggregation is a first step towards comparing inversion estimates with NGHGIs, as inversions may have difficulties to disentangle overlapping emissions. Removing the natural emissions is an important step for the comparison.

Line 670: missing units on the vertical axes

The unit is labeled in the first plot in Fig 3.

Line 744: FAO estimates for india CH4 agriculture are much closer to the inversions than those reported in the NGHGI. Same

is true for bangladesh

Thanks. Comparing FAO estimates might be beyond the scope of our paper. Note that this shows that NGHGI can be quite different/dependent than other global inventories used as prior (FAO, EDGAR, etc).

Line 787: according to which dataset

We calculate the percentage contributions according to the median of inversion data we used in this studys.

Line 791: considering that n most countries agriculture emits close to 75% of all N2O, have you considered checking FAOSTAT data for some coomparisons

Thanks. Comparing FAO estimates might be beyond the scope of our paper.

Line 791: why were these data not disaggreagted by sector as done for teh CH4

The inverse models cannot separate as many sectors as CH4.

Line 801: can you quantify the impact? it does not seem likely that these N2O emissions can compare in scale to those due to agriculture and hence have any impact in this comparison

In our study, the national N2O emissions drop around 15% in these two countries after masking the flux from intact forests.

Line 809: that may be, but the IPCC EFs come with huge uncertainty of +- 75-100%, so including this could perhaps reduce the apparent discrepancies or at least help address the comparison, within the limitations of the underlying uncertainties in both NGHGI and inversion model results.

Thanks. We revise the sentence to: "In practice, the EFs are mostly based on measurements made in temperate climates and for soils of cropland established long ago with little 'background' emissions, so there may be a systematic underestimation of default IPCC EFs of emissions from tropical climates and recently established agricultural land when the IPCC EFs also has huge uncertainty to ±75-100%"

Line 819: did you check the data with the actual data owners, to investigate uncertainty and limitations in their use for this exercise

No, comparing FAO estimates might be beyond the scope of our paper.

Line 847: considering that a good amount of fossil fuel emissions happens in OECD countries plus China (which reports to UNFCCC), it would appear that the excellent agreement between inversions forced with priors that are close to the average of the NGHGI data is not such an exceptional fact. Kindly add a sentence that better explains this to the reader, possibly with a quantitative example.

We discussed it as: "These emissions are not optimized by inversions and may differ from UNFCCC NGHGIs because: most non-Annex I countries do not have annual emissions estimates and inversions use as fixed priors gap-filled annual data from CDIAC and BP statistics for non-Annex I countries from Friedlingstein et al. (2020), whereas the red circumferences in Fig 8 show the average of emissions from available BUR or NC reports in each period (thus, for each period only data of available years of national inventories are used to calculate the average). For Annex I countries, the prior fossil CO2 emission maps prescribed to the inversions from Jones et al. (2021) match by construction to national totals from UNFCCC. For non-Annex I countries, the prior fossil CO2 emission map is based mainly on CDIAC national emissions (Friedlingstein et al., 2020)."

Line 850: bit this is an aggregate of AI countries --the sentence before states that possible mismatches are likely only for NAI parties. Perhaps it's misleading to start this sentence with "therefore". Kindly edit accordingly.

Thanks. We revised this sentence: "The data presented in Fig 8 show differences between NGHGIs data and inversion priors, possibly due to:"

Line 853: wouldn't these three factors apply across the board, not only to EUR?

Yes. EUR is just an example.

Line 856: there is something akin to circularity in this apparently. Inversions are based on priors, which are based on international databases that in turn are based at least in part on the very NGHGI that are used to assess the "goodness" of the exercise. There should be a substantial discussion on this at the outset, in the methods section.

Thanks. We added discussions in Section 1.1: "We subtracted the same fossil fuel emissions from Friedlingstein et al. (2020) from the total CO2 flux of each inversion to analyze terrestrial CO2 fluxes, which is equivalent to assuming perfect knowledge of emissions but note that these values are consistent with the fossil fuel emissions reported in the NGHGIs. This assumption

leads to an under-estimation of the spread of terrestrial CO2 fluxes among inversions."

Line 859: is this small or not? If the exercise is to use inversions results to perform QAQC on NGHGI, or even further along, to report directly GHG emissions reductions, 37% seems like a lot of error to monitor reductions that are likely to be in the next decade in the order of 0-30% of current emissions

Compared with 0-30%, 37% is undoubtedly large. However, this study is only an attempt to give a quantitative estimate of the difference between the two.

Line 861: this is an interesting take on the analysis. The authors seem to start by thinking that the inversions are right and the NGHGI are wrong, and proceed to explore reasons why the NGHGI could be wrong. However there are at the outset a number of definitional issues and uncertainties that need to enter this discussion at the start, or the risk is that all these arguments are interesting and intellingently made, but rest substantially on very thin air.

Thanks for the comment. To avoid possibly biased language, we revised a few sentences in the manuscript and replaced "under-reporting" which conveys the idea of a deliberate under-reporting by "lower estimation".

Line 862: yes, but would would this increase the observed sink? At best, the world grasslands are being heavily degraded according to UNCCD, so why would such processes lead to a carbon sink. Secondly, adding sink capacity from the soil component under forests, which is typically ignored in inventories, would likely not be sufficient to correct for the huge differences discussed for some countries. For forests in particular, the inversions results seem to go teh opposite directions from data provided not only to UNFCCC, but also in terms of carbon stock and area info submitted to the FAO and which have been shown to result overall in a net source--once deforestation and peatland degradation processes are factored in.

We acknowledge that more research is needed to understand better WHERE is the sink "seen" by inversions but not reported in NGHGIs. This is a key point for the GST, but to my knowledge no paper provided really clear answer to that

Line 867: this is also true for independent estimates based on national level carbon stock change made by FAO recently.

Thanks.

Line 871: has this been confirmed with Australian national NGHGI compilers? or is it in the NIR? A reference is needed. Same for CAN

Yes, it's confirmed by the compilers through personal communications.

Line 874: but the inversions results already point to much larger sinks in these areas -- likely these sinks would be larger over the entire unmasked area!

The area of unmanaged land we defined in this study by using the intact forests might be larger than the unmanaged land defined in the NGHGIs, thus leading to a larger sinks in these regions.

Line 877: the mismatch is due to the fact that managed/unmanaged land definitions appear to be different between this community and the IPCC guidelines --more discussion of this should be provided at the outset. What the authors call a "political" decision is in fact a difference of interpretation of the statistical concept of "land use" -- which doe snot only cover land for economic production, but indeed also most administrative arrangements set up by countries to prescribe uses of land --including protection, tribal land etc. To this end, most countries' lands are in fact managed. The use of the intact forest mask is somewhat arbitrary --and introduced additional uncertainty.

Thanks. We add discussions of the different managed/unmanaged land definitions.

Line 883: regardless, one could compare the independent estimates by FAO, which computes carbon fluxes over the entire forest area-- for comparison and quantitative insight into the importance or not of this issue.

Thanks. Comparing FAO estimates might be beyond the scope of our paper. Note that this shows that NGHGI can be quite different/dependent than other global inventories used as prior (FAO, EDGAR, etc).

Line 884: why not? one could use the FAO gridded livestock map of the world to do just that.

Thanks for the suggestion. The definition of managed grasslands is different with livestock maps which will introduce further uncertainties and differences into this comparison. It could be further discussed in future studies.

Line 887: first, definitions again. These are not "grasslands" in teh land cover sense. THey are "grasslands" in the IPCC land use sense, hence, accoridng to FAO definitions "permanent meadows and pastures" including rangelands and all sorts of savannah and tundra ecosystems.

Thanks. Here we use the definition of 'grasslands' in IPCC.

Line 907: definition?

We have defined it in section 1.3: "Over a country that receives carbon from rivers flowing into its territory, a small national $CO_2$ outgassing is produced by a fraction of this imported flux. In that case, we assumed that the fraction of outgassed to incoming river carbon is equal to the fraction of outgassed to soil-leached carbon in the RECCAP2 region to which a country belongs to, estimated with data from Ciais et al. (2020b)."

Line 919: here we are in for lots of uncertainty and again crossed definitions. Countries report to FAO and FAO gap gills to produce a database on land use where the terms are cropland +permanent meadows and pasture = agricultural land, + forest land. These are land uses, which should be used to produce information that compares to NGHGI data. Conversely, land cover information is i) not land use; and ii) extremely dependent on the land cover product. MODIS, CCI, GLC give entirely different results in many parts of the world for these (mapped) land use categories (and differ even on the land cover info).
Line 921: not sure I understand this. The actual source of sink will depend on actual land amounts (hectares), not on a relative ratio
Line 930: somehow this all sounds like speculation with little additional analysis conducted to argue for or against a certain hypothesis. It's fine but please frame these sentences accordingly to fully convey this is speculation. What would be needed to improve on this analysis considering teh high uncertainty in the underlying results?
Line 933: how does this work exactly? don't coarser resolutions work either way potentially? Whether a country is a source or a sink depends form many biophysical and management factors beyond resolution.
Line 937: how would the correlation work simply against total area of forest land? would it not make more sense to regress against it rather than based on relative factors--which do not tell anything about the state of the forest nor its area extension. I would take this section out altogether

Thanks. We removed this section.

Line 950: the graph shows exceptional agreement, in particular in depicting the right scales by country. How much is this a result of using priors and successive calibration? IN other words, how independent are the inversions results

Please refer to our reply for the comment #1

Line 990: there are alternative data in FAOSTAT that could be used directly as primers for anthropogenic biomass burning.
Line 1041: considering that agriculture is roughly 80% of N2O anthropogenic emissions, why did you not use FAOSTAT as primer data? All countries emissions are estimated 1961-2019.

Thanks. Comparing FAO estimates might be beyond the scope of our paper. Note that this shows that NGHGI can be quite different/dependent than other global inventories used as prior (FAO, EDGAR, etc).

Line 1053: unlikely to be an issue in terms of anthropogenic emissions, as the lion's share of N inputs into managed soils is on croplands.

The anthropogenic emissions are calculated based on activity data and from the managed land. So it would be a important source of uncertainty when we define the areas of unmanaged land.

Line 1057: ok, it may be. But again, the authors seem to systematically assume that the inversions are more correct than the data in the NGHGIs. What could instead have gone wrong in the inversions?

We are giving the possible reason for narrowing the gap between inversions and inventories, without any assumption that inversions or inventories are more accurate. In fact, the inversions may have larger uncertainties than the NGHGIs. However, there are still some improvements in inventory compilation to narrow the gap. We revised this sentence to make it clear: "The consistent pattern of higher emissions in inversions than NGHGIs among the three model inversions for non-Annex I countries, suggests the possible improvements in inventory compilation including adopting country-specific emission factors , or re-assessing and reporting indirect emissions when it is not the case (Table S2)."